# DATS: Distance-Aware Temperature Scaling for Calibrated Class-Incremental Learning

## Abstract

Continual Learning (CL) is recently gaining increasing attention for its ability to enable a single model to learn incrementally from a sequence of new classes. In this scenario, it is important to keep consistent predictive performance across all the classes and prevent the so-called Catastrophic Forgetting (CF). However, in safety-critical applications, predictive performance alone is insufficient. Predictive models should also be able to reliably communicate their uncertainty in a calibrated manner – that is, with confidence scores aligned to the true frequencies of target events. Existing approaches in CL address calibration primarily from a data-centric perspective, relying on a *single* temperature shared across all tasks. Such solutions overlook task-specific differences, leading to large fluctuations in calibration error across tasks. For this reason, we argue that a more principled approach should adapt the temperature according to the distance to the current task. However, the unavailability of the task information at test time/during deployment poses a major challenge to achieve the intended objective. For this, we propose Distance-Aware Temperature Scaling (DATS), which combines prototype-based distance estimation with distance-aware calibration to infer task proximity and assign adaptive temperatures without prior task information. Through extensive empirical evaluation on both standard benchmarks and real-world, imbalanced datasets taken from the biomedical domain, our approach demonstrates to be stable, reliable and consistent in reducing calibration error across tasks compared to state-of-the-art approaches.

## 1 Introduction

The steep improvement of the predictive capabilities of modern neural architectures has led practitioners to increasingly deploy neural networks into critical decision-making systems. In these contexts, however, predictive models must not only be accurate but also calibrated – i.e., able to reliably communicate via uncertainty estimates when they are likely to be incorrect. Yet, despite their accuracy, neural networks often show under- or over-confidence, especially under distribution shifts. Model calibration offers a way to ensure that a model's predicted confidence levels are statistically consistent with the empirical frequency of correct predictions. For instance, among all predictions assigned a confidence level of 85%, the model should yield correct outputs in approximatively 85% of the cases. Consequently, model calibration has become progressively more investigated over the years to enhance the reliability and trustworthiness of neural architectures in standard single-task settings (Guo et al., 2017; Zhang et al., 2020; Gupta et al., 2021; Tomani et al., 2022).

However, in many real-world applications – ranging from predicting virus variants to product recommendation in e-commerce –, the environment is not static but may vary over time. For this reason, Continual Learning (CL) has gained increasing attention for its ability to enable a single model to learn incrementally from a sequence of new classes. In this scenario, due to the tendency of the model to focus on the most recent information, it is important to keep consistent predictive performance across all the classes and thus mitigate the so-called Catastrophic Forgetting (CF) (McCloskey & Cohen, 1989; Ratcliff, 1990). Reducing the forgetting effect in CL has been largely investigated in recent years (Lopez-Paz & Ranzato, 2017; Chaudhry et al., 2019; Tao et al., 2020; Bang et al., 2021; Hurtado et al., 2023; Serra et al., 2025) but remains a major challenge yet to be completely solved.

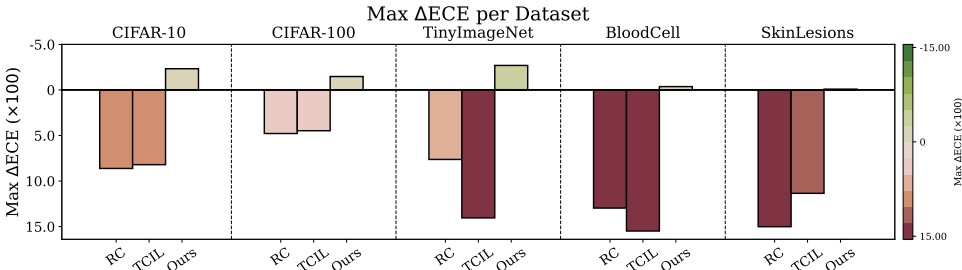

Figure 1: Comparison of the worst-case difference in Expected Calibration Error (Max $\Delta$ECE) after post-hoc re-calibration. Positive values (red) indicate worsened calibration (higher CE than before re-calibration), while negative ones (green) indicate improved calibration (lower calibration error than before re-calibration). We observe that state-of-the-art methods can substantially increase calibration error for certain tasks, whereas our approach consistently reduces miscalibration.

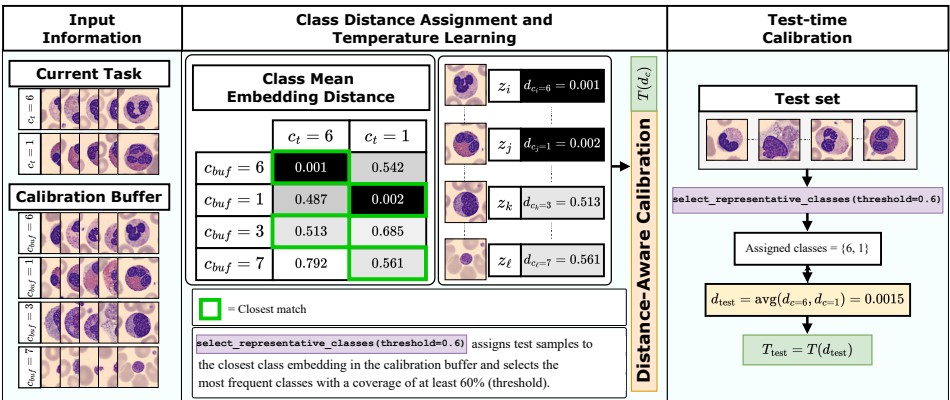

Figure 2: Schematic overview of the proposed framework. Given the validation set of the current task $D_{t,val}$ and the calibration buffer $\mathcal{B}_t$, our method first aligns each class in $\mathcal{B}_t$ (i.e., $c_{buf}$) with the closest class in $D_{t,val}$ (i.e., $c_t$). Each sample in $\mathcal{B}_t$ (e.g., $x_i$) is then assigned a distance score $d_{c_{buf}=c_i}$. During calibration, both the logit vector $z_i$ and the corresponding distance score serve as inputs to the distance-aware optimisation phase. At test time, samples from $D_{t,test}$ are mapped to their nearest counterparts in $\mathcal{B}_t$; the assigned classes provide a reference for computing $d_{test}$ and guide the final distance-aware temperature scaling.

Considering the forgetting effect and the dynamic characteristics of the setting, CL models are intrinsically more prone to make wrong predictions than single-task models. Thus, it is even more important to ensure that CL models reliably communicate their uncertainty. Let's consider a practical example as a running example throughout this work; in one hospital, a neural network is trained continuously to detect skin lesions. The model will initially be trained on the most common types of skin lesions (e.g., melanocytic nevi) but, as more data are collected, new and rarer classes (e.g., vascular lesions, dermatofibroma) will be encountered during training. For the deployment of such model in critical scenarios, the model is required to be accurate but, more importantly, should be able to reliably communicate its uncertainty such that human intervention can be requested when complex cases are encountered. For such uncertainty estimates to be useful in practice, they need to be calibrated. In standard single-task classification scenarios, model calibration can be substantially improved by employing post-hoc uncertainty calibration approaches (Platt et al., 1999; Zadrozny & Elkan, 2002; Zhang et al., 2020; Tomani et al., 2023) - that is, after training model outputs are transformed, most commonly by dividing logits by a learnt temperature (temperature scaling), such that they better match the true likelihood.

Despite the relevance of the problem, post-hoc uncertainty calibration, is largely under-explored in CL. The limited literature available (Li et al., 2024; Hwang et al., 2025) tackles the problem from a data-centric perspective by learning a *single* temperature shared across *all* the involved tasks. How-

ever, due to catastrophic forgetting, predictive performance often differs substantially between tasks, indicating that such task-agnostic strategies fail to capture the dynamic nature of CL problems. This motivates us to shift towards *task-aware* calibration methods in order to account for the variability across tasks.

In an ideal setting, each task could be re-calibrated by using its own validation set. Yet, this is impractical during deployment – and in class-incremental learning (CIL) – where the task information is not available at any time. Thus, to solve this challenge and introduce task-awareness into the re-calibration process, we propose Distance-Aware Temperature Scaling (DATS), a method that a) infers the proximity to the current task via a prototype-based distance criterion and b) exploits such information to inject task-awareness during calibration optimisation and, in the context of temperature-based approaches, learns how to assign a temperature based on the estimated distance. This mitigates miscalibration due to temperature variability across tasks.

The contributions of this work can be summarised as follows:

- We demonstrate that, given the dynamic characteristics of CL (Figure 3), calibration should move from task-agnostic to *task-aware* and *adaptive* re-calibration methods and expose the limitations of current task-agnostic data-centric approaches (Figure 4).
- We propose a distance-aware calibration framework that shifts the focus from data selection and injects task-awareness directly into the re-calibration algorithm. Our approach allows us to assign a distance score to each test batch, enabling *task-aware* re-calibration without explicit task information.
- We show the effectiveness of our method on standard benchmarks and more probing real-world scenarios from the medical domain.

## 2 PRELIMINARIES AND RELATED WORK

**Preliminaries**   Following the notation proposed in Hwang et al. (2025), we assume to have a dataset $D_t = \{(x_i, y_i)\}$ for each task $t$, such that each class label $y_i \in D_t$ belongs to a disjoint set of classes $C_t$. The dataset is split in $D_{t,train}$, $D_{t,val}$, and $D_{t,test}$. We also assume to have a memory buffer $\mathcal{M}$, which stores a subset of samples from the previously encountered tasks. We denote with $C$ a set of classes and with $\mathcal{C}$ the union of set of classes. Thus, let $\mathcal{C}_{t-1} = \bigcup_{k=1}^{t-1} C_k$ denote the set of all previously seen classes up to task $t-1$; then, the memory $\mathcal{M}_{t-1}$ contains a limited set of labelled samples $(x_j, y_j)$ such that $y_j \in \mathcal{C}_{t-1}$. The model is trained on $D_t$ and uses $\mathcal{M}_{t-1}$ for replay. At the end of the training of task $t$, the memory $\mathcal{M}_t$ is updated with a portion of samples taken from $D_{t,train}$. Finally, following Li et al. (2024), we also assume to have a calibration buffer $\mathcal{B}_t$ which contains a subset of samples from the validation sets encountered up to task $t$.

Let $f_\theta$ be a classifier parametrised by $\theta$, producing a logit vector $z_i = f_\theta(x_i)$ for an input $x_i$. The function maps input data to $K$ output classes, where $K$ is the size of $\mathcal{C}_t$ (i.e., the total number of classes at time $t$). The probability vector $p_i$ is obtained by passing the logits $z_i$ through a softmax function such that $p_i = \mathrm{softmax}(z_i)$. Then, we denote with $\hat{y}_i = \arg\max_k p_{i,k}$ and $\hat{p}_i = \max_k p_{i,k}$ the predicted class and the confidence of the prediction respectively.

Using the notion of calibration, we can define *perfect calibration* (Guo et al., 2017) as:

$$\mathbb{P}(\hat{y}_i = y_i | \hat{p}_i = p) = p \quad \forall p \in [0, 1]. \tag{1}$$

It naturally follows the definition of Calibration Error (CE) (Naeini et al., 2015) which measures the gap between predicted confidence and actual accuracy:

$$\mathbb{E}_{(x_i, y_i) \sim \mathcal{P}} \left[ |\mathbb{P}(\hat{y}_i = y_i \mid \hat{p}_i) - \hat{p}_i| \right]. \tag{2}$$

However, in practical scenarios with a finite number of samples, the probability defined in Equation 2 cannot be computed exactly, and binning-based estimators are commonly used. Let us divide the interval $[0, 1]$ in $B$ equally-spaced bins $b$. For each bin $B_b$, we can compute the empirical average accuracy and confidence via $\mathrm{acc}_b = \frac{1}{|B_b|} \sum_{i \in b} \mathbb{1}(\hat{y}_i = y_i)$ and $\mathrm{conf}_b = \frac{1}{|B_b|} \sum_{i \in b} \hat{p}_i$ respectively. Finally, we can estimate the expected calibration error (ECE) via

$$\mathrm{ECE} = \sum_{b=1}^{B} \frac{|B_b|}{N} |\mathrm{acc}_b - \mathrm{conf}_b|, \tag{3}$$

where $N$ represents the total number of samples in the considered dataset $D$.

**Post-hoc calibration methods**  Among existing calibration strategies, post-hoc methods have gained particular popularity due to their ease of application. Unlike approaches that require modifying the classifier's training procedure (Müller et al., 2019; Moon et al., 2020; Ghosh et al., 2022; Liu et al., 2022; Noh et al., 2023), post-hoc calibration is applied after training, making it flexible and computationally efficient. Several post-hoc techniques have been proposed, including Platt Scaling (Platt et al., 1999), Isotonic Regression (IR) (Zadrozny & Elkan, 2002), and Spline Calibration (Gupta et al., 2021). Following prior work in continual learning (Li et al., 2024; Hwang et al., 2025), we focus on temperature scaling (TS, Guo et al. (2017)). TS learns a single scalar parameter $T$ on a validation set to rescale the classifier's logits: predictions are softened if overconfident ($T > 1$) or sharpened if underconfident ($T < 1$). The method is order-preserving and efficient, as it requires learning only one parameter. More advanced variants of TS extend this idea, for example through ensembling (ETS, Zhang et al. (2020)) or parameterised temperature scaling (PTS, Tomani et al. (2022)), where $T$ is modelled sample-wise by a multi-layer perceptron (MLP). In all these standard TS approaches, calibration relies only on the logit vector $z_i$ of each sample. Taking inspiration from the literature in out-of-distribution detection, where Sun et al. (2022) introduce a non-parametric density estimation based on k-nearest neighbour (KNN) distance, Tomani et al. (2023) propose to exploit the information contained in the inner layers of the classifier (up to the penultimate one) as an additional input for KNN. This density-based information is then used, together with $z$, to learn sample-wise temperatures for uncertainty calibration in OOD scenarios.

**Calibration in CL**  The challenge of post-hoc uncertainty calibration in CL has only been recently investigated. Li et al. (2024) present the first study of this problem in class-incremental learning (CIL) in the context of memory-based approaches. The authors introduce Replayed Calibration (RC), an approach that leverages an additional buffer (namely, a calibration buffer) populated with samples taken from the validation set of each task for calibration optimisation. Despite its simplicity, this work opens a new research direction and highlights the importance of exploiting information from both current and past tasks for calibration. More recently, Hwang et al. (2025) propose T-CIL, a new temperature scaling (TS) approach for CIL. Starting from the observation that using the memory buffer for temperature optimisation is not effective due to its usage during training, the authors propose to perturb the samples contained in the memory buffer to create synthetic samples for calibration. In particular, the magnitude and the direction of the perturbations is adjusted based on the difficulty of the samples taken into consideration – samples from old tasks are perturbed more strongly than the current task's ones.

RC and T-CIL primarily emphasise the choice of the data used for calibration rather than the calibration mechanism itself. For instance, T-CIL generates synthetic samples for calibration starting from the memory buffer and the current validation set, but its generative nature introduces substantial computational overhead and significantly increases the execution time (see Table 15). Ultimately, both methods rely on the same principle of learning a single temperature across tasks while avoiding direct use of the memory buffer. Such task-agnostic approach, however, fails to account for task-specific variability and thus appears insufficient to mitigate miscalibration in CL. In contrast, we depart from this task-agnostic and data-centric perspective (i.e., which data to use for calibration) and are the first to explicitly consider the dynamic nature of CL, introducing task-awareness into post-hoc calibration to better account for the variations across tasks.

## 3 METHODOLOGY

As argued above, the evolving nature of CL motivates the need for a *task-aware* re-calibration approach. Catastrophic forgetting causes predictive performance to vary substantially across tasks, and we hypothesize that this discrepancy is reflected in the confidence scores of the model. We investigate this behaviour in Figure 3, which reports the average confidence score per task for two different datasets. In both cases, the most recent task exhibit significantly higher confidence than earlier ones. This observation supports our hypothesis: although accuracy for the last task is typically higher, a *task-agnostic* approach cannot yield well-calibrated predictions across all tasks.

To make this point explicit, we translate the *task-agnostic* vs. *task-aware* dilemma in the context of temperature scaling. In a synthetic experiment (Figure 4), we compute the ideal temperature per task by using its corresponding validation set. The results clearly show that the optimal temperature

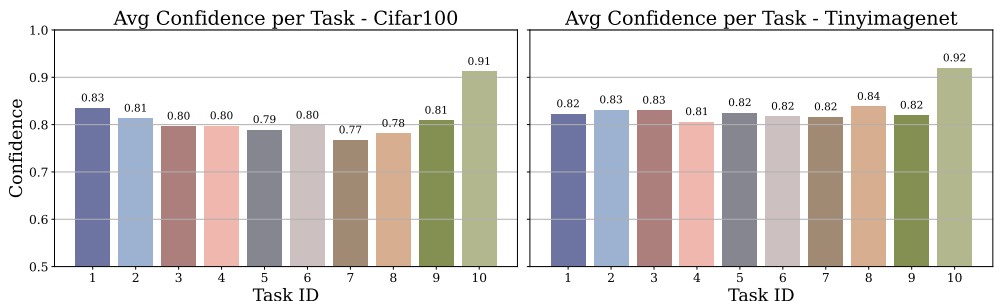

Figure 3: Visualisation of the average confidence score for each task at the end of the training procedure. A clear difference is noticeable between the average confidence of the last task and the one of the past tasks.

per task fluctuates across past tasks and decreases for the most recent ones, demonstrating that a *single* temperature approach is *not sufficient* for effective calibration in CL.

These findings motivate us to move beyond task-agnostic strategies and develop *adaptive*, *task-aware* post-hoc calibration methods. In the context of temperature scaling, this means moving from learning a *single* temperature for all tasks to learning *different* temperatures depending on the task at hand. In this way, we can instil task-awareness into the re-calibration phase and overcome the limitations of current task-agnostic state-of-the-art approaches. In the ideal scenario presented in Figure 4, one could calibrate each task with its validation set, but this is impractical in class-incremental learning and during deployment where task labels are not available at any time. To address this challenge, we propose to infer the task information indirectly through prototype-based distance information and use this signal to tune the temperature accordingly.

In the remainder of this section, we describe in more details the modules constituting our approach. First, in Section 3.1, we describe how to compute and assign distance scores to classes in the calibration buffer. Then, in Section 3.2, we illustrate the use of the learned scores for task-aware temperature optimisation procedure. Finally, in Section 3.3, we outline the strategy to assign a distance score to the test set without task knowledge and to adapt the temperature accordingly. A summary of the whole method is described in Appendix A.9 - Algorithm 1.

## 3.1 CLASS DISTANCE SCORE ASSIGNMENT

The main objective in this phase is to identify the closest class pairs between the current classes (contained in $D_{t,val}$) and those in the calibration buffer (i.e., all seen classes). For this purpose, let $C_t$ be the set of classes for the current task $t$, and $\mathcal{C}_{buf} = \mathcal{C}_t = \bigcup_{k=1}^{t} C_k$ be the set of buffer classes containing all classes seen up to and including task $t$.

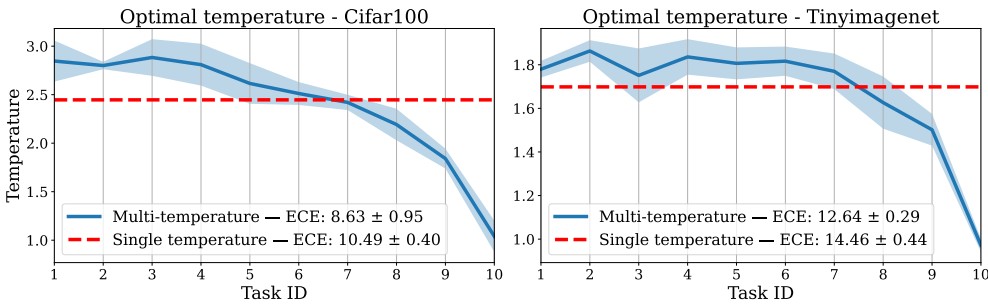

Figure 4: Comparison between the ideal temperature for each task learned by exploiting the corresponding validation set $D_{t,val}$ (blue line) and the optimal single-temperature learned by concatenating the validation information of all tasks (red dashed line).

To quantify the relationship between classes in the current task and those stored in the calibration buffer, we represent each class by a prototype embedding $\boldsymbol{\mu}_c$, computed as the average of the latent representations at the penultimate layer $\ell - 1$ of the neural network:

$$\boldsymbol{\mu}_c = \frac{1}{|D_c|} \sum_{(x_i, y_i) \in D_c} f_\theta^{\ell-1}(x_i), \tag{4}$$

where $D_c$ denotes the set of samples belonging to class $c$, and $f_\theta^{\ell-1}(x_i)$ is the output of the penultimate layer for input $x_i$. For the current task classes, these prototypes are computed from $D_{t,val}$, while for buffer classes they are computed from the samples stored in $\mathcal{B}_t$.

Then, to assess the similarity between buffer and validation classes, we compute the pairwise cosine similarity matrix $\boldsymbol{S} \in \mathbb{R}^{|C_t| \times |\mathcal{C}_{buf}|}$:

$$\boldsymbol{S}(c_t, c_{buf}) = \frac{\boldsymbol{\mu}_{c_t}^\top \boldsymbol{\mu}_{c_{buf}}}{\|\boldsymbol{\mu}_{c_t}\|_2 \, \|\boldsymbol{\mu}_{c_{buf}}\|_2}, \quad c_t \in C_t, \ c_{buf} \in \mathcal{C}_{buf}. \tag{5}$$

Finally, the score for each buffer class $c_{buf}$ is defined as the minimum distance (based on cosine similarity) to any current class:

$$d(c_{buf}) = \min_{c_t \in C_t} \left(1 - \boldsymbol{S}(c_t, c_{buf})\right) \quad \forall c_{buf} \in \mathcal{C}_{buf}. \tag{6}$$

The scores are then normalised via MinMaxScaler. Intuitively, we expect $d(c_{buf}) \approx 0$ when $c_{buf} \in C_t$, and $d(c_{buf}) \gg 0$, otherwise. The score is assigned back to the examples in the buffer such that the ones from the same class $c$ share the same score $d(c_{buf} = c)$. An illustration of this assignment is depicted in Figure 2; the learned distance scores are assigned back to samples in $\mathcal{B}_t$ according to the corresponding class.

## 3.2 TASK-AWARE CALIBRATION

Once we assign a distance-per-class score $d(c_{buf})$ to each sample in the calibration buffer $\mathcal{B}_t$, we leverage these scores as an additional signal during the calibration optimisation phase. Here, $d(c_{buf})$ acts as a proxy for the distance between a buffer sample and the current task, thereby introducing task-awareness into the calibration process. While the logit information $\boldsymbol{z}$ already provide fine-grained, sample-specific information, we assign the same distance score to all samples of a given class $c$. This choice reduces the information burden during temperature optimisation and encourages the model to capture class-level temperature dynamics. In contrast, using per-sample distances would introduce unnecessary noise and instability, since the variability within a class may impede learning the broader class-level trends that are most relevant for inducing *task-aware* calibration.

For a given class $c$, the temperature is defined as

$$T(d_c) = w_c d_c + T_{base}, \tag{7}$$

where $T_{base}$ is the global base temperature and $w_c$ controls how strongly the distance score $d_c$ modulates the adjustment. In this way, our method can learn a temperature for each seen class (i.e., for each class $c \in \mathcal{C}_{buf}$). We learn $T_{base}$ and $w_c$ by minimising the Brier score based on the logits, distance scores and labels from $\mathcal{B}_t$ (see Equation 9). More details about the optimisation of our task-aware re-calibration approach are reported in Appendix A.1.

## 3.3 TEST-TIME CALIBRATION

At test time, our goal is to apply $T(d_c)$ to $D_{t,test}$ for all tasks up to $t$. Since class and task information are not available in this phase, we proceed as follows. First, each test sample is assigned to the nearest class embedding $\boldsymbol{\mu}_c$ in the calibration buffer $\mathcal{B}_t$. Then, we retain only the most frequent classes covering at least $60\%$ of the assignments. This filtering step ensures that the selected classes reliably describe the test set while reducing the risk of including spurious classes. For this, considering the catastrophic forgetting effect, we adopt a $60\%$ threshold as a practical trade-off between coverage and representativeness of each task: higher thresholds increase the likelihood of incorporating misclassified classes, thereby amplifying discrepancies between assigned and true classes (see Appendix A.8, Table 16).

Through this class assignment, we approximate the dominant classes in the test set and use them to compute a representative score for the entire set. Specifically, given the set of assigned classes $\hat{C}_{\text{test}}$ and the distance score obtained from the calibration buffer, we define the test set distance $d_{\text{test}}$ as

$$d_{\text{test}} = \frac{1}{|\hat{C}_{\text{test}}|} \sum_{c \in \hat{C}_{\text{test}}} d(c_{buf} = c). \tag{8}$$

A numerical example of this procedure is illustrated in Figure 2. Suppose we are processing the test set of the current task with classes 6 and 1 and, with a threshold of 0.6, the class assignment step successfully retrieves the two classes. Then, $d_{\text{test}}$ is computed via Equation 8 and used as input to calculate the temperature according to Equation 7. Intuitively, when the class assignment is (almost) correct, we expect to have $d_{\text{test}} \approx 0$ when dealing with the current task and $d_{\text{test}} > 0$ otherwise.

## 4 EXPERIMENTS

### 4.1 DATASETS AND SETTINGS

**Datasets** In line with related work (Li et al., 2024; Hwang et al., 2025), we evaluate our method on three popular datasets: CIFAR10, CIFAR100 (Krizhevsky et al., 2009), and TinyImageNet (Le & Yang, 2015). To reproduce the class-incremental scenario, we divide each dataset into disjoint tasks. For CIFAR10, we randomly assign two classes to each task (5 tasks). For CIFAR100 and TinyImageNet, we respectively assign 10 and 20 classes to every task (10 tasks). This setup allows the evaluation of baselines independently of task composition. As anticipated in Section 1, we next evaluate our approach under more challenging and realistic settings. Instead of using CIFAR-like datasets (e.g., ImageNet) or artificial tasks (e.g., EMNIST), we conduct validation on biomedical image analysis datasets. Beyond domain differences in image statistics, these datasets also introduce an additional challenge: class imbalance. To better imitate realistic scenarios, where newer tasks typically provide fewer samples due to limited collection time or due to class rarity, we assign classes to tasks according to their relative sizes. This setting represents a realistic and common scenario in AI-assisted medicine where one hospital may encounter different pathology subtypes with different frequencies. For this purpose, we focus on two biomedical datasets, both annotated by expert clinical pathologists; BloodCell (Acevedo et al., 2020; Yang et al., 2021), with 8 classes of microscopic blood cell images, and SkinLesions (Tschandl et al., 2018; Codella et al., 2019; Yang et al., 2021), with 7 classes of dermoscopic images for skin cancer detection. Statistics are reported in Appendix A.4.

**Experimental settings** In all the main experiments, we use a slim version of ResNet18 (He et al., 2016) as done in previous CL works (Lopez-Paz & Ranzato, 2017; Kumari et al., 2022; Hurtado et al., 2023; Serra et al., 2025) and use the SGD optimizer with a learning rate of 0.1. For replay, we use Experience Replay (ER) (Chaudhry et al., 2019). In order to achieve competitive results on each dataset and have a meaningful baseline, the size of the memory buffer $\mathcal{M}$ is adapted according to the dataset in consideration. It is important to notice that, since our approach is post-hoc, the initial training procedure can be changed as desired as long as the trained model achieves reasonable predictive results. We ablate the use of a different architecture in Appendix A.8 - Table 18. For the composition of the calibration buffer $\mathcal{B}$, following Li et al. (2024), we reserve with random selection a percentage of the validation set $D_{t,val}$ of each processed task. We ablate the percentage value in Appendix A.8 - Table 17. A detailed list of the hyperparameter selection can be found in the supplementary material. We run the experiments on three random seeds such that each time the class-per-task assignment is different. Each experiment was run on a Linux machine using a single Quadro RTX 5000 with 16 GB RAM.

**Baselines** In line with prior work (Li et al., 2024; Hwang et al., 2025), we first benchmark our method against standard temperature scaling (TS) techniques (i.e., TS, Ensemble TS (ETS), and Parametrised TS (PTS)) which rely only on the current validation set. This comparison highlights the inefficacy of approaches that ignore information from previous tasks in CL. Then, we compare DATS against RC (Li et al., 2024) and T-CIL (Hwang et al., 2025) to assess the effectiveness of our approach in comparison with task-agnostic approaches specifically tailored for class-incremental learning (CIL).

Table 1: Comparison of average negative log-likelihood (NLL), average expected calibration error (ECE) and average delta last ECE (ΔLECE) on standard benchmarks.

| | CIFAR10 (Acc: 63.58 ± 2.86) | | | CIFAR100 (Acc: 45.19 ± 0.89) | | | TinyImageNet (Acc: 22.79 ± 0.38) | | |
|---|---|---|---|---|---|---|---|---|---|
| | NLL (▼) | AECE (▼) | ΔLECE (▼) | NLL (▼) | AECE (▼) | ΔLECE (▼) | NLL (▼) | AECE (▼) | ΔLECE (▼) |
| Uncal | 3.38 ± 0.54 | 30.77 ± 3.01 | ✗ | 4.02 ± 0.07 | 35.55 ± 0.47 | ✗ | 4.27 ± 0.13 | 38.09 ± 1.23 | ✗ |
| TS | 2.33 ± 0.28 | 27.94 ± 2.95 | -1.14 ± 0.25 | 4.03 ± 0.59 | 35.38 ± 2.64 | 0.08 ± 1.93 | 4.36 ± 0.09 | 39.45 ± 0.50 | 0.55 ± 0.35 |
| ETS | 2.31 ± 0.34 | 28.26 ± 2.75 | -1.00 ± 0.18 | 3.90 ± 0.60 | 35.67 ± 2.00 | 0.18 ± 1.62 | 4.36 ± 0.09 | 39.45 ± 0.51 | 0.55 ± 0.35 |
| PTS | ✗ | ✗ | ✗ | 5.96 ± 1.50 | 34.92 ± 2.06 | 1.48 ± 2.34 | 5.10 ± 1.08 | 37.50 ± 2.83 | -0.09 ± 0.31 |
| RC | 1.12 ± 0.06 | 12.18 ± 2.33 | 7.68 ± 1.24 | 2.32 ± 0.07 | 10.03 ± 0.32 | 6.44 ± 5.98 | 3.52 ± 0.04 | 14.45 ± 0.59 | 7.24 ± 2.87 |
| T-CIL | 1.12 ± 0.06 | 12.33 ± 2.68 | 7.34 ± 2.01 | 2.32 ± 0.08 | 9.56 ± 1.29 | 7.11 ± 3.70 | 3.52 ± 0.02 | **10.48 ± 0.78** | 16.77 ± 3.84 |
| Ours | **1.08 ± 0.04** | **8.80 ± 0.93** | **-2.31 ± 0.75** | **2.29 ± 0.06** | **7.63 ± 0.84** | **-1.26 ± 1.21** | **3.50 ± 0.02** | 11.84 ± 0.58 | **-2.66 ± 1.67** |

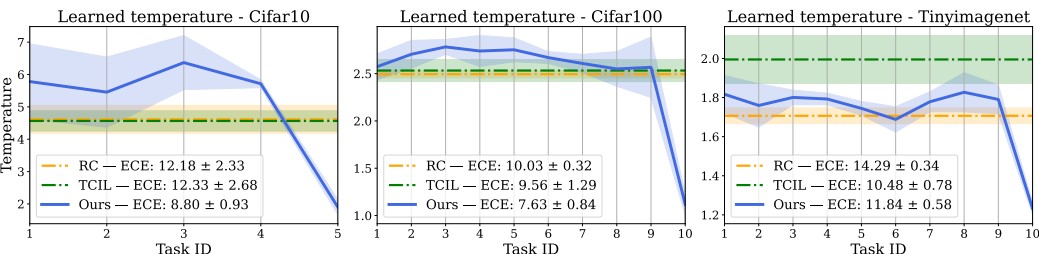

Figure 5: Visual comparison of the learned temperatures across tasks between task-agnostic (RC and T-CIL) and task-aware (ours) calibration methods for CL.

**Evaluation metrics** Following standard practice in CIL settings, we report the *average* metric. For instance, the average accuracy (*Acc* in our tables) report the average of the accuracy across tasks at the end of the training procedure. Similarly, we report the average negative log-likelihood (NLL) and the average expected calibration error (AECE). We adopt both ECE and NLL as complementary measures of calibration quality. While NLL assesses the overall quality of the predictive distribution, ECE directly measures the alignment between predicted confidence and empirical accuracy. Furthermore, we introduce two additional metrics to evaluate calibration in CL settings: the difference in ECE before and after calibrating the last task (ΔLECE) and the worst-case difference in ECE across tasks after re-calibration (max ΔECE). We propose ΔLECE to show the effect of the adopted temperature on the current task; considering the change in the confidence distribution between the current and previous tasks, this metric gives an idea of the effect of the calibration on the currently evaluated task. Instead, max ΔECE provides insight into the stability and robustness of a calibration method. For both metrics, negative values mean that the considered approach effectively reduces the calibration error, while positive values indicate increased ECE. A detailed definition of the metrics is provided in Appendix A.6.

## 4.2 EMPIRICAL RESULTS

From the results reported in Table 1, we can observe that the proposed approach consistently reduces miscalibration in most cases in terms of NLL and ECE. Looking at the difference between the calibration error before and after calibrating the last task (ΔLECE), it is evident that existing solutions produce an undesirable behaviour: miscalibration becomes even more severe than when no re-calibration is applied. The learned temperatures shown in Figure 5 further illustrate these dynamics, where it is clear that a task-agnostic fixed temperature does not capture the dynamic properties of CIL settings. Noticeably, while RC and T-CIL perform similarly on CIFAR datasets, T-CIL assign a much higher temperature on TinyImageNet. This results in a smaller average ECE compared to our method, but at the cost of a substantially larger calibration error on the last task. Importantly, we also demonstrate that our approach remains effective and stable on imbalanced, real-world datasets from the biomedical domain, as reported in Table 2.

In practical deployments, and especially in critical domains as our running example described in Section 1, our method proves safer and more reliable by avoiding large fluctuations across tasks and

Table 2: Comparison of average negative log-likelihood (NLL), average expected calibration error (ECE) and average delta last ECE (ΔLECE) on BloodCell and SkinLesions.

| | Blood Cell (Acc: 77.90 ± 4.46) | | | Skin Lesions (Acc: 49.14 ± 0.67) | | |
|---|---|---|---|---|---|---|
| | NLL (▼) | AECE (▼) | ΔLECE (▼) | NLL (▼) | AECE (▼) | ΔLECE (▼) |
| Uncal | 1.25 ± 0.24 | 16.47 ± 3.54 | ✗ | 1.57 ± 0.08 | 16.48 ± 0.88 | ✗ |
| RC | 0.82 ± 0.12 | 9.91 ± 1.04 | 12.53 ± 1.63 | 1.44 ± 0.02 | 14.26 ± 1.58 | 14.77 ± 2.31 |
| T-CIL | 0.83 ± 0.11 | 10.08 ± 1.24 | 13.31 ± 2.74 | 1.43 ± 0.03 | 13.09 ± 0.76 | 9.56 ± 1.56 |
| Ours | **0.80** ± **0.11** | **9.87** ± **1.40** | **0.21** ± **0.66** | **1.42** ± **0.09** | **12.53** ± **2.68** | **-1.90** ± **1.81** |

preventing severe degradation on individual ones. This is illustrated in Figure 1, where we show the worst-case ECE improvement ($\max \Delta$ECE) across tasks for each baseline. A positive delta (in red) indicates that the ECE of a given task is increased after calibration, while a negative delta (in green) represents decreased ECE. From the results, we can observe that, in the worst case, our approach does not change or marginally improves the calibration of a task, while current state-of-the-art approaches exacerbate the problem in all the considered datasets. This property is particularly important in safety-critical settings, such as biomedical applications, where rare or newly introduced classes are at risk of being severely miscalibrated by approaches like RC or T-CIL. In such cases, our method offers a more stable and trustworthy calibration strategy across tasks.

## 5 DISCUSSION AND CONCLUSION

**Execution time** Due to space constraints, we report the runtime of the calibration phase in seconds in Appendix A.8 - Table 15. The results show that, while DATS introduces some overhead compared to RC, it remains competitively fast compared to RC and considerably faster than T-CIL.

**Comprehensive evaluation tailored to the CL setting** Beyond standard metrics, we introduce two new evaluation measures: 1) $\Delta$LECE to assess the impact of calibration on the most recently learned task (given the change in its predictive performance and confidence distribution), and 2) $\max \Delta$ECE which captures whether re-calibration degrades calibration on any task by tracking the worst-case change in ECE across the sequence. In line with our shift from a task-agnostic to a task-aware re-calibration perspective, these metrics provide a more nuanced understanding of re-calibration effectiveness beyond average metrics.

**Ablation studies** In Appendix A.8, we analyse the impact of the most relevant hyperparameters on the performance of our method. From the results (Tables 16, 17, and 18), our approach remains stable and robust within a broad range of settings.

**Limitations** The approach assumes that the representative classes selected at test-time are a reliable proxy of the processed task. However, under highly non-stationary streams, this assumption may be weakened.

**Conclusions** In this work, we tackle the challenges of uncertainty calibration in CL, a largely under-explored problem. By investigating the behaviour of CL models (Figure 3), we argue that post-hoc calibration techniques in CL should move from a task-agnostic to a task-aware perspective. Current state-of-the-art approaches, primarily focusing on the data to use for calibration, overlook task variability and propose a task-agnostic single temperature approach which appears to be limited in CL settings (Figure 4). This choice, in fact, leads to large fluctuations in ECE across tasks (Figure 1), an undesirable behaviour for the deployment of CL models in safety-critical settings. For this reason, we depart from such task-agnostic, data-centric view and are the first to propose an adaptive, task-aware calibration method for CL. To integrate task-awareness into the calibration phase, we propose DATS, an elegant solution that, in the context of TS, is able to adapt the task temperature without prior information about it via a combination of prototype-based distance estimation and distance-aware calibration. The empirical results show that our approach delivers robust calibration across tasks on a variety of datasets, ranging from standard benchmark (Table 1) to real-world, im-balanced biomedical ones (Table 2). Additional ablation studies demonstrates that DATS is simple to tune and efficient, providing an easy-to-integrate tool for post-hoc recalibration of CL models. Overall, DATS offers a lightweight, principled, and effective solution for *task-aware calibration* in class-incremental learning, enabling safer deployment of CL models in safety-critical domains.

REPRODUCIBILITY STATEMENT

**Code availability.** We provide the full implementation of our method, along with all scripts necessary to reproduce the experiments. The code is submitted as supplementary material and will be made publicly available upon acceptance.

**Data accessibility.** All datasets employed in our experiments are publicly available. We include detailed instructions on dataset usage, and any required preprocessing steps are automated via scripts included in the code repository.

**Hyperparameters.** Complete training configurations and hyperparameter values are specified in the main text. The effect of critical hyperparameters is further examined in the ablation study (see Appendix A.8).

**Hardware and runtime.** Experiments were conducted on a Linux server with a single NVIDIA Quadro RTX 5000 GPU and 16 GB RAM. Training times are reported in Table 15.

**Experiment instructions.** The repository includes a README file with step-by-step instructions for reproducing the results. For clarity, we also provide pseudocode for the main components of our method in Appendix A.9.

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

# A  APPENDIX

## A.1  POST-HOC CALIBRATION OPTIMISATION

Given the logits $z$, the distance scores $d(c_{\text{buf}})$, and the labels from the calibration buffer $\mathcal{B}_t$, we fit our task-aware, class-specific temperature scaling parameters for the trained classifier $f_\theta$ by minimizing the Brier score using the L-BFGS optimizer:

$$L_{\text{cal}} = \sum_{i=1}^{N} \sum_{c \in \mathcal{C}_{buf}} \left( I_{i,c} - \text{softmax}((z_i^c / T(d_c)) \right)^2 \tag{9}$$

where $N$ is the number of samples in $\mathcal{B}_t$, $I_{i,c}$ is an indicator variable equal to 1 if $y_i = c$ and 0 otherwise, and $T(d_c)$ denotes the temperature assigned to class $c$ as a function of its distance score $d_c$.

## A.2  THEORETICAL JUSTIFICATION FOR PROTOTYPE-BASED DISTANCES

In this section, we provide theoretical justifications for using class mean prototypes as distance measures in DATS. While class prototypes do not fully characterize the underlying feature distribution and cannot capture higher-order statistics, our method does not require full distributional modelling. Below, we outline the theoretical assumptions under which prototype-based distances are justified for our approach, along with their connection to temperature scaling and task recency.

**Class means are theoretically sufficient under standard feature-space assumptions.** Many analyses of deep representation spaces (e.g., Mahalanobis-based OOD detectors (Lee et al., 2018)) assume that penultimate-layer features are locally Gaussian with *approximately shared within-class covariance* across tasks:

$$h \mid (y = c, t) \sim \mathcal{N}(\mu_{t,c}, \Sigma), \quad \Sigma_{t,c} \approx \Sigma. \tag{10}$$

Under this shared-covariance model, all standard distributional discrepancies between two class-conditional distributions (e.g., KL divergence, Mahalanobis distance, and Wasserstein-2) reduce to functions of the mean difference:

$$\mathrm{KL}\left(\mathcal{N}(\mu_{t,c}, \Sigma) \,\|\, \mathcal{N}(\mu_{t',c}, \Sigma)\right) = \tfrac{1}{2}(\mu_{t,c} - \mu_{t',c})^\top \Sigma^{-1}(\mu_{t,c} - \mu_{t',c}). \tag{11}$$

Thus, the class mean is a sufficient statistic for quantifying inter-task drift at the class level in this setting. Our prototype is simply the empirical estimator of this mean, and our cosine-based distance serves as a practical surrogate for such Mahalanobis-style divergences.

Furthermore, class-incremental learning can be viewed as a form of temporal distribution shift, where old task classes become "out-of-distribution" relative to the most recent training regime. Even when the Gaussian shared-covariance assumption does not hold exactly, using penultimate layer features has been demonstrated to work well in OOD scenarios since the penultimate layer preserves relevant information for the task (Sun et al., 2022).

**Perfect alignment with temperature scaling capacity.** Temperature scaling adjusts only a single scalar parameter. It can correct global logit-scale mismatches but cannot account for differences in covariance or higher-order statistics. Therefore, using first-order statistics (i.e., class means) is not a limitation but matches exactly the expressive capacity of temperature scaling. Incorporating higher-order statistics would provide information that TS cannot exploit.

Furthermore, the corresponding estimates would be highly unstable given the limited-size calibration buffer. The calibration buffer $\mathcal{B}_t$ typically contains very few samples per class (e.g., $N = 20$) in a high-dimensional feature space (e.g., $D = 512$):

- For the estimation of the mean ($\mu$), the error rate converges at $O(\sqrt{D/N})$.
- For the estimation of the covariance ($\Sigma$), the error rate is significantly higher, requiring $O(D^2)$ parameters. With $N \ll D$, the sample covariance matrix becomes singular and ill-conditioned, making reliable estimation infeasible.

Thus, while prototypes do not encode the entire class distribution, they provide exactly (i) the theoretically justified first-order statistics that dominate inter-task drift under shared-covariance models, and (ii) the level of information that temperature scaling can meaningfully exploit. This makes prototype distances an appropriate and principled choice for our calibration framework. Furthermore, the effectiveness of our method across multiple architectures, continual learning baselines, and datasets provides strong empirical evidence that prototype distances are the primary driver of calibration errors in class-incremental learning.

## A.3 DATS$_w$: A THRESHOLD-FREE ALTERNATIVE

In some cases, it might be convenient to not rely on any threshold selection and let our approach decide how to compute $d_{\text{test}}$. For this, to not depend on the threshold hyperparameter, we present a weighted strategy denoted with DATS$_w$. We can replace Equation 8 with:

$$d_{\text{test}} = \sum_{c \in \hat{C}_{\text{test}}} \frac{n_c}{|D_{\text{t,test}}|} d(c_{\text{buf}} = c), \tag{12}$$

where $n_c$ represent the number of samples in $D_{\text{t,test}}$ assigned to class $c \in \hat{C}_{\text{test}}$. In other words, Equation 12 represents a weighted average of the assigned distance scores. The results of DATS$_w$ are comparable to DATS on all the considered datasets and strategies, demonstrating its effectiveness while avoiding tuning the threshold explicitly.

Table 3: Comparison of average negative log-likelihood (NLL), average expected calibration error (ECE) and average delta last ECE ($\Delta$LECE) between DATS and DATS$_w$ on standard benchmarks.

| | CIFAR10 (Acc: $63.58 \pm 2.86$) | | | CIFAR100 (Acc: $45.19 \pm 0.89$) | | | TinyImageNet (Acc: $22.79 \pm 0.38$) | | |
|---|---|---|---|---|---|---|---|---|---|
| | NLL (▼) | AECE (▼) | $\Delta$LECE (▼) | NLL (▼) | AECE (▼) | $\Delta$LECE (▼) | NLL (▼) | AECE (▼) | $\Delta$LECE (▼) |
| Uncal | $3.38 \pm 0.54$ | $30.77 \pm 3.01$ | ✗ | $4.02 \pm 0.07$ | $35.55 \pm 0.47$ | ✗ | $4.27 \pm 0.13$ | $38.09 \pm 1.23$ | ✗ |
| DATS | $\mathbf{1.08} \pm \mathbf{0.04}$ | $\mathbf{8.80} \pm \mathbf{0.93}$ | $-2.31 \pm 0.75$ | $2.29 \pm 0.06$ | $7.63 \pm 0.84$ | $-1.26 \pm 1.21$ | $3.50 \pm 0.02$ | $11.84 \pm 0.58$ | $\mathbf{-2.66} \pm \mathbf{1.67}$ |
| DATS$_w$ | $1.09 \pm 0.04$ | $9.16 \pm 1.28$ | $\mathbf{-3.34} \pm \mathbf{1.22}$ | $\mathbf{2.28} \pm \mathbf{0.06}$ | $\mathbf{7.04} \pm \mathbf{0.78}$ | $\mathbf{-6.03} \pm \mathbf{0.41}$ | $\mathbf{3.49} \pm \mathbf{0.02}$ | $\mathbf{10.88} \pm \mathbf{0.31}$ | $-1.39 \pm 1.63$ |

Table 4: Comparison of average negative log-likelihood (NLL), average expected calibration error (ECE) and average delta last ECE ($\Delta$LECE) between DATS and DATS$_w$ on BloodCell and SkinLesions.

| | Blood Cell (Acc: $77.90 \pm 4.46$) | | | Skin Lesions (Acc: $49.14 \pm 0.67$) | | |
|---|---|---|---|---|---|---|
| | NLL (▼) | AECE (▼) | $\Delta$LECE (▼) | NLL (▼) | AECE (▼) | $\Delta$LECE (▼) |
| Uncal | $1.25 \pm 0.24$ | $16.47 \pm 3.54$ | ✗ | $1.57 \pm 0.08$ | $16.48 \pm 0.88$ | ✗ |
| DATS | $\mathbf{0.80} \pm \mathbf{0.11}$ | $9.87 \pm 1.40$ | $0.21 \pm 0.66$ | $1.42 \pm 0.09$ | $12.53 \pm 2.68$ | $\mathbf{-1.90} \pm \mathbf{1.81}$ |
| DATS$_w$ | $0.81 \pm 0.12$ | $\mathbf{9.54} \pm \mathbf{1.87}$ | $\mathbf{0.10} \pm \mathbf{0.54}$ | $\mathbf{1.39} \pm \mathbf{0.04}$ | $\mathbf{10.70} \pm \mathbf{1.41}$ | $0.09 \pm 2.19$ |

## A.4 DATASET STATISTICS

In Table 5, we report the statistics of the datasets used for the main experiments. Following related work, we use standard benchmarks (CIFAR10, CIFAR100, TinyImagenet), and datasets from different domains (SkinLesions and BloodCell). In particular, the two biomedical datasets pose additional challenges as they reflect more realistic conditions (less data, imbalanced).

Table 5: Statistics of the datasets.

| Number of | Classes | Samples | Tasks | Image size |
|---|---|---|---|---|
| CIFAR10 | 10 | 60000 | 5 | 32 |
| CIFAR100 | 100 | 60000 | 10 | 32 |
| TinyImageNet | 200 | 100000 | 10 | 64 |
| SkinLesions | 7 | 10015 | 3 | 64 |
| BloodCell | 8 | 17092 | 4 | 28 |

## A.5 HYPERPARAMETER SELECTION

In Table 6, we report the hyperparameters used for each dataset. The choices follow standard practice in CL literature to obtain a backbone configuration that achieves reasonable predictive performance. Specifically, NF denotes the number of features in the ResNet backbone, $|\mathcal{M}|$ is the memory buffer size, Patience controls early stopping during training, and and Val. inclusion (%) indicates the percentage of each task's validation set that is included in the calibration buffer for calibration optimisation. Values for Val. inclusion (%) were chosen per dataset to reflect dataset size and class balance; small datasets or imbalanced biomedical datasets use larger fractions to ensure sufficient calibration samples. For standard benchmarks, the value is chosen to match the size of the memory buffer. An ablation study for the percentage value is reported in Table 17.

Table 6: Hyperparameter selection for each dataset.

| | NF | $|\mathcal{M}|$ | Patience | Val. inclusion (%) |
|---|---|---|---|---|
| CIFAR10 | 20 | 1000 | 10 | 10 |
| CIFAR100 | 32 | 4000 | 20 | 40 |
| TinyImageNet | 64 | 1000 | 50 | 20 |
| SkinLesions | 64 | 200 | 20 | 50 |
| BloodCell | 20 | 200 | 50 | 50 |

## A.6 EVALUATION METRICS

We adopt a set of standard metrics for CL together with additional measures specifically designed to assess calibration in class-incremental learning (CIL) settings. Following standard practice, we report the *average* values of the considered metrics across tasks at the end of training. In particular:

- Accuracy (*Acc*) evaluates predictive performance in terms of correct classifications.
- Negative Log-Likelihood (*NLL*) and Expected Calibration Error (*ECE*) serve as complementary measures of calibration quality: NLL captures the overall quality of the predictive distribution, while ECE directly measures the alignment between predicted confidence and empirical accuracy.
- $\Delta$LECE and $\max \Delta$ECE are additional metrics we introduce to assess the effect and robustness of calibration in CL settings.

A detailed definition of each metric follows.

- *Average Accuracy (Acc)*: Let $\text{acc}^{t,i}$ denote the accuracy on task $i$ after learning task $t$. Given the total number of tasks $\mathcal{T}$, the average accuracy $Acc$ is defined as:

$$Acc = \frac{1}{\mathcal{T}} \sum_{i=1}^{\mathcal{T}} \text{acc}^{\mathcal{T},i}. \tag{13}$$

  This metric summarizes the predictive performance across all tasks at the end of the training procedure.

- *Average Negative Log-Likelihood (NLL)*: The negative log-likelihood measures the quality of the full predictive distribution. Lower values indicate that the predicted probabilities are well aligned with the true labels, penalizing both overconfidence and underconfidence. Let $\text{nll}^{t,i}$ denote the negative log-likelihood on task $i$ after learning task $t$. The average NLL is defined as:

$$NLL = \frac{1}{\mathcal{T}} \sum_{i=1}^{\mathcal{T}} \text{nll}^{\mathcal{T},i}. \tag{14}$$

- *Average Expected Calibration Error (AECE)*: The expected calibration error measures the discrepancy between predicted confidence and empirical accuracy, typically estimated via binning. Lower values correspond to better-calibrated models. Let $\text{ece}^{t,i}$ denote the ECE on task $i$ after learning task $t$ computed via Equation 3 with $B = 10$. The average ECE (AECE) is defined as:

$$AECE = \frac{1}{\mathcal{T}} \sum_{i=1}^{\mathcal{T}} \text{ece}^{\mathcal{T},i}. \tag{15}$$

- *Delta Last ECE ($\Delta$LECE)*: To assess the effect of the adopted calibration procedure on the last task, we define the change in last-task ECE. Let $\text{ece}^{t,t}$ and $\text{ece-cal}^{t,t}$ denote the ECE on task $t$ before and after calibration, respectively. Then:

$$\Delta\text{LECE} = \text{ece-cal}^{\mathcal{T},\mathcal{T}} - \text{ece}^{\mathcal{T},\mathcal{T}}. \tag{16}$$

  This metric quantifies the direct impact of calibration on the most recently learned task.

- *Worst-case Delta ECE ($\max \Delta$ECE)*: To evaluate the stability and robustness of a calibration method across tasks, we consider the worst-case change in ECE. Let $\Delta\text{ece}^{\mathcal{T},i} = \text{ece-cal}^{\mathcal{T},i} - \text{ece}^{\mathcal{T},i}$ be the change in ECE for task $i$ after calibration at the end of training. The worst-case delta ECE is defined as:

$$\max \Delta\text{ECE} = \max_{i \in \{1,\dots,\mathcal{T}\}} \Delta\text{ece}^{\mathcal{T},i}. \tag{17}$$

  This metric captures the most adverse calibration effect across tasks, highlighting whether re-calibration destabilizes the calibration performance across tasks.

For both $\Delta$LECE and $\max \Delta$ECE, negative values indicate that the considered approach reduces miscalibration (improved calibration), while positive values indicate an increase in miscalibration.

## A.7 Additional Results with Other CL Baselines

To demonstrate broader applicability of our approach and its plug-in nature, we perform additional experiments considering different families of CL strategies. In particular, we decided to employ the following strategies: GDumb (Prabhu et al., 2020) (bias-free), Learning without Forgetting (LwF) (knowledge distillation) (Li & Hoiem, 2017), Elastic Weight Consolidation (EWC) (parameter regularisation) (Kirkpatrick et al., 2017), and Retrospective Adversarial Replay (RAR) (more sophisticated memory-based approach) (Kumari et al., 2022). Since miscalibration stems primarily from the classifier being biased toward the most recent tasks, GDumb represents an ideal experiment to validate the benefit of using DATS in settings where this bias is naturally reduced. DATS still consistently reduces calibration error on top of GDumb across all datasets. This demonstrates that our task-aware temperature scaling approach provides benefits that go beyond what can be achieved through bias mitigation alone. Furthermore, the results on additional baselines demonstrate that our approach can be combined with diverse CL strategies to consistently reduce ECE in diverse settings – including low-accuracy regimes – while maintaining the simplicity and plug-and-play nature that makes it practically valuable.

### A.7.1 GDumb

Table 7: Calibration performance when using GDumb during training. Results are consistent with those obtained using Experience Replay (ER), confirming that our approach is not sensitive to the choice of the employed CL strategy.

| | CIFAR10 (Acc: 33.35 ± 3.25) | | | CIFAR100 (Acc: 24.27 ± 0.57) | | | TinyImageNet (Acc: 10.59 ± 1.59) | | |
|---|---|---|---|---|---|---|---|---|---|
| | NLL (▼) | AECE (▼) | ΔLECE (▼) | NLL (▼) | AECE (▼) | ΔLECE (▼) | NLL (▼) | AECE (▼) | ΔLECE (▼) |
| GDumb | 1.91 ± 0.06 | 14.71 ± 2.44 | ✗ | 3.42 ± 0.11 | 15.03 ± 2.75 | ✗ | 4.38 ± 0.08 | 5.29 ± 0.48 | ✗ |
| +DATS | **1.83** ± **0.05** | 12.68 ± 2.04 | **1.61** ± **0.91** | **3.16** ± **0.03** | 3.79 ± 0.10 | -7.56 ± 5.46 | **4.29** ± **0.10** | 2.96 ± 0.16 | -1.97 ± 3.39 |
| +DATS$_w$ | 1.83 ± 0.06 | **12.54** ± **2.17** | 3.42 ± 2.01 | **3.16** ± **0.03** | 3.81 ± 0.19 | -7.46 ± 5.39 | **4.29** ± **0.10** | **2.93** ± **0.16** | -2.02 ± 3.34 |

Table 8: Calibration performance when using GDumb during training on BloodCell and SkinLesions. Results are consistent with those obtained using ER.

| | Blood Cell (Acc: 68.94 ± 13.57) | | | Skin Lesions (Acc: 46.95 ± 4.02) | | |
|---|---|---|---|---|---|---|
| | NLL (▼) | AECE (▼) | ΔLECE (▼) | NLL (▼) | AECE (▼) | ΔLECE (▼) |
| GDumb | 1.36 ± 0.76 | 19.65 ± 10.99 | ✗ | 1.50 ± 0.25 | 15.97 ± 6.70 | ✗ |
| +DATS | **0.85** ± **0.36** | **8.85** ± **5.43** | **-1.10** ± **3.42** | 1.44 ± 0.10 | 14.97 ± 3.73 | 0.47 ± 9.30 |
| +DATS$_w$ | **0.85** ± **0.36** | 8.99 ± 5.02 | 4.85 ± 0.94 | **1.42** ± **0.13** | **13.36** ± **5.24** | **-0.47** ± **12.06** |

### A.7.2 Learning without Forgetting (LwF)

Table 9: Calibration performance when using Learning without Forgetting (LwF) during training. Results are consistent with those obtained using Experience Replay (ER), confirming that our approach is not sensitive to the choice of the employed CL strategy.

| | CIFAR10 (Acc: 62.19 ± 2.40) | | | CIFAR100 (Acc: 45.13 ± 0.63) | | | TinyImageNet (Acc: 11.89 ± 0.61) | | |
|---|---|---|---|---|---|---|---|---|---|
| | NLL (▼) | AECE (▼) | ΔLECE (▼) | NLL (▼) | AECE (▼) | ΔLECE (▼) | NLL (▼) | AECE (▼) | ΔLECE (▼) |
| LwF | 3.06 ± 0.40 | 28.87 ± 2.65 | ✗ | 2.99 ± 0.09 | 27.38 ± 0.84 | ✗ | 10.74 ± 0.21 | 57.33 ± 1.41 | ✗ |
| +DATS | 1.29 ± 0.07 | 9.83 ± 4.73 | -0.76 ± 1.12 | **2.26** ± **0.03** | 6.36 ± 0.21 | -1.06 ± 0.48 | **4.58** ± **0.03** | **8.98** ± **0.28** | 1.69 ± 6.79 |
| +DATS$_w$ | **1.29** ± **0.06** | **9.20** ± **4.22** | **-2.21** ± **0.72** | **2.26** ± **0.03** | **6.29** ± **0.21** | **-3.04** ± **0.47** | 4.61 ± 0.03 | 10.20 ± 0.55 | 5.24 ± 1.52 |

Table 10: Calibration performance when using Learning without Forgetting (LwF) during training on BloodCell and SkinLesions. Results are consistent with those obtained using ER.

| | Blood Cell (Acc: 80.05 ± 1.63) | | | Skin Lesions (Acc: 47.05 ± 6.23) | | |
|---|---|---|---|---|---|---|
| | NLL (▼) | AECE (▼) | ΔLECE (▼) | NLL (▼) | AECE (▼) | ΔLECE (▼) |
| LwF | 1.09 ± 0.05 | 13.96 ± 1.26 | ✗ | 1.46 ± 0.09 | 15.56 ± 4.12 | ✗ |
| +DATS | 0.89 ± 0.07 | **11.50** ± **1.37** | 0.70 ± 0.28 | 1.40 ± 0.09 | 13.22 ± 4.71 | **-1.21** ± **1.00** |
| +DATS$_w$ | **0.83** ± **0.13** | 12.19 ± 1.22 | **0.69** ± **0.28** | **1.38** ± **0.08** | **11.06** ± **2.52** | 0.96 ± 2.91 |

### A.7.3 ELASTIC WEIGHT CONSOLIDATION (EWC)

Table 11: Calibration performance when using Elastic Weight Consolidation (EWC) during training. Results are consistent with those obtained using Experience Replay (ER), confirming that our approach is not sensitive to the choice of the employed CL strategy.

| | CIFAR10 (Acc: 64.72 ± 2.26) | | | CIFAR100 (Acc: 48.23 ± 0.77) | | | TinyImageNet (Acc: 10.79 ± 0.45) | | |
|---|---|---|---|---|---|---|---|---|---|
| | NLL (▼) | AECE (▼) | ΔLECE (▼) | NLL (▼) | AECE (▼) | ΔLECE (▼) | NLL (▼) | AECE (▼) | ΔLECE (▼) |
| EWC | 2.07 ± 0.35 | 27.08 ± 2.81 | ✗ | 2.63 ± 0.05 | 27.56 ± 0.84 | ✗ | 6.11 ± 0.13 | 54.41 ± 0.93 | ✗ |
| +DATS | **1.12** ± **0.05** | **8.05** ± **1.84** | -1.12 ± 2.72 | 2.05 ± 0.02 | 7.92 ± 0.56 | -0.82 ± 1.66 | **4.28** ± **0.03** | 12.63 ± 0.46 | **-4.53** ± **0.82** |
| +DATS$_w$ | 1.13 ± 0.05 | 9.65 ± 2.05 | **-1.35** ± **2.85** | **2.04** ± **0.02** | **7.88** ± **0.60** | **-3.13** ± **0.23** | 4.28 ± 0.02 | **12.44** ± **0.22** | 1.31 ± 0.90 |

Table 12: Calibration performance when using Elastic Weight Consolidation (EWC) during training on BloodCell and SkinLesions. Results are consistent with those obtained using ER.

| | Blood Cell (Acc: 78.63 ± 1.79) | | | Skin Lesions (Acc: 48.83 ± 2.67) | | |
|---|---|---|---|---|---|---|
| | NLL (▼) | AECE (▼) | ΔLECE (▼) | NLL (▼) | AECE (▼) | ΔLECE (▼) |
| EWC | 1.25 ± 0.17 | 15.61 ± 1.68 | ✗ | 1.55 ± 0.11 | 15.04 ± 2.49 | ✗ |
| +DATS | 0.84 ± 0.21 | 9.28 ± 2.15 | 0.57 ± 0.68 | 1.42 ± 0.04 | 11.55 ± 3.02 | **2.35** ± **3.74** |
| +DATS$_w$ | **0.79** ± **0.20** | **8.70** ± **1.85** | **0.56** ± **0.67** | **1.41** ± **0.05** | **10.28** ± **1.69** | 2.84 ± 4.48 |

### A.7.4 RETROSPECTIVE ADVERSARIAL REPLAY (RAR)

Table 13: Calibration performance when using Retrospective Adversarial Replay (RAR) during training. Results are consistent with those obtained using Experience Replay (ER), confirming that our approach is not sensitive to the choice of the employed CL strategy.

| | CIFAR10 (Acc: 63.05 ± 1.08) | | | CIFAR100 (Acc: 40.73 ± 1.15) | | | TinyImageNet (Acc: 20.53 ± 1.45) | | |
|---|---|---|---|---|---|---|---|---|---|
| | NLL (▼) | AECE (▼) | ΔLECE (▼) | NLL (▼) | AECE (▼) | ΔLECE (▼) | NLL (▼) | AECE (▼) | ΔLECE (▼) |
| RAR | 1.66 ± 0.12 | 24.48 ± 1.94 | ✗ | 2.90 ± 0.05 | 24.84 ± 0.82 | ✗ | | | ✗ |
| +DATS | **1.08** ± **0.02** | **7.31** ± **0.64** | -0.53 ± 1.04 | **2.45** ± **0.04** | 6.89 ± 0.07 | -1.64 ± 2.38 | | | |
| +DATS$_w$ | **1.08** ± **0.02** | 8.15 ± 0.38 | **-1.66** ± **1.93** | **2.45** ± **0.04** | **6.27** ± **0.36** | **-4.10** ± **1.43** | | | |

Table 14: Calibration performance when using Retrospective Adversarial Replay (RAR) during training on BloodCell and SkinLesions. Results are consistent with those obtained using ER.

| | Blood Cell (Acc: 35.77 ± 12.00) | | | Skin Lesions (Acc: 45.46 ± 0.64) | | |
|---|---|---|---|---|---|---|
| | NLL (▼) | AECE (▼) | ΔLECE (▼) | NLL (▼) | AECE (▼) | ΔLECE (▼) |
| RAR | 4.42 ± 1.73 | 46.39 ± 10.79 | ✗ | 2.00 ± 0.03 | 26.05 ± 8.65 | ✗ |
| +DATS | **1.59** ± **0.33** | **16.45** ± **6.81** | **-4.64** ± **4.71** | **1.54** ± **0.13** | 18.51 ± 1.69 | -10.08 ± 8.68 |
| +DATS$_w$ | 1.62 ± 0.35 | 17.34 ± 5.78 | -3.92 ± 4.28 | **1.54** ± **0.13** | **18.38** ± **1.51** | **-10.42** ± **9.16** |

## A.8 ABLATION STUDY

In this section, we ablate different experimental choices to investigate the effectiveness and robustness of our approach to different settings.

**Execution time** Table 15 reports the runtime of the calibration phase in seconds. As expected, RC represents the lower bound since it simply applies TS on a calibration buffer without introducing additional computation. Despite the extra steps required by our method, DATS remains competitively fast compared to RC and considerably faster than T-CIL. Notably, the generative nature of T-CIL induces substantial computational overhead leading to longer execution times across datasets.

Table 15: Execution time (in seconds) of the calibration phase after training. After the first task, the value is the sum of the calibration procedure for each task up to the current one.

| | CIFAR10 | CIFAR100 | TinyImageNet | Blood Cell | Skin Lesions |
|---|---|---|---|---|---|
| | Runtime ($\blacktriangledown$, in seconds) | | | | |
| RC | $19.09 \pm 2.06$ | $23.76 \pm 0.12$ | $376.02 \pm 18.11$ | $3.37 \pm 0.04$ | $3.92 \pm 0.00$ |
| T-CIL | $230.88 \pm 1.46$ | $510.80 \pm 21.49$ | $1949.55 \pm 55.38$ | $38.52 \pm 0.16$ | $59.15 \pm 0.36$ |
| Ours | $22.15 \pm 0.12$ | $54.36 \pm 7.87$ | $479.61 \pm 14.03$ | $7.50 \pm 0.17$ | $9.88 \pm 0.04$ |

**Coverage threshold** As described in Section 3.3, at test time, we assign test samples to the closest class embedding $\mu_c$ in the calibration buffer $\mathcal{B}$. Considering that CL models are more prone to be inaccurate (especially for past tasks), we decide to set the coverage threshold to 0.6 for all the experiments as a good compromise between representativeness and coverage. In Table 16, we ablate the coverage threshold (i.e, the percentage of samples in the test set that are used to infer the representative classes). From the results, we can notice a clear trend; reducing the threshold improves the final ECE in most of the cases. This effect is expected since, by reducing the threshold, we are filtering out the least present classes and, most probably, the mismatch between the real and assigned classes on the test set. This is particularly true for more challenging datasets like CIFAR100 and TinyImageNet, where the overall accuracy is less than $50\%$. In general, we believe a value between $40\%$ and $60\%$ to be a good range for most of the cases.

Table 16: Effect of varying the coverage threshold on the calibration performance (ECE $\blacktriangledown$). Best results are typically obtained for thresholds between 40% and 60%

| Threshold | 0.4 | 0.5 | 0.6 | 0.7 | 0.8 |
|---|---|---|---|---|---|
| | ECE ($\blacktriangledown$) | | | | |
| CIFAR10 | $9.53 \pm 0.60$ | $8.80 \pm 0.93$ | $8.80 \pm 0.93$ | $9.40 \pm 1.80$ | $12.49 \pm 2.040$ |
| CIFAR100 | $6.52 \pm 0.62$ | $6.73 \pm 0.49$ | $7.63 \pm 0.84$ | $8.38 \pm 0.99$ | $8.70 \pm 0.35$ |
| TinyImageNet | $10.31 \pm 0.32$ | $11.29 \pm 0.48$ | $11.84 \pm 0.58$ | $11.97 \pm 0.40$ | $12.32 \pm 0.39$ |

**Validation inclusion percentage** In Table 17, we analyse the effect of the size of the calibration buffer $\mathcal{B}$ in terms of the percentage of each task's validation set $D_{t,val}$ included in $\mathcal{B}$ (Val. inclusion (%)). For each validation set, samples are drawn uniformly at random. From the results, the percentage of samples included from $D_{t,val}$ has only a minor impact on the calibration performance of our approach.

Table 17: Effect of varying the percentage of each task's validation set (Val. inclusion (%)) on the calibration performance (ECE $\blacktriangledown$). Results in terms of ECE do not change considerably when changing the percentage of samples selected from $D_{t,val}$.

| Val. inclusion (%) | 20 | 30 | 40 | 50 |
|---|---|---|---|---|
| | ECE ($\blacktriangledown$) | | | |
| CIFAR10 | $8.85 \pm 0.81$ | $8.99 \pm 0.68$ | $9.02 \pm 0.67$ | $8.83 \pm 0.75$ |
| CIFAR100 | $7.50 \pm 0.55$ | $7.44 \pm 0.95$ | $7.63 \pm 0.84$ | $7.81 \pm 0.67$ |
| TinyImageNet | $11.84 \pm 0.58$ | $11.56 \pm 0.85$ | $11.67 \pm 0.81$ | $11.46 \pm 0.82$ |

**Backbone architecture**  In our experiments, we use a slim version of ResNet18 (SlimResNet) for all the dataset. In Table 18, we report the results on the benchmark datasets when using a different architecture for training, i.e., ResNet32 (He et al., 2016). We can see that changing the backbone architecture does not affect the capabilities of our approach and the behaviour of all the considered methods.

Table 18: Calibration performance when using ResNet32 as backbone architecture. Results are consistent with those obtained using SlimResNet, confirming that our approach is not sensitive to the choice of the backbone architecture.

| | CIFAR10 (Acc: $65.43 \pm 2.09$) | | | CIFAR100 (Acc: $44.81 \pm 2.43$) | | | TinyImageNet (Acc: $20.53 \pm 1.45$) | | |
|---|---|---|---|---|---|---|---|---|---|
| | NLL (▼) | ECE (▼) | ΔLECE (▼) | NLL (▼) | ECE (▼) | ΔLECE (▼) | NLL (▼) | ECE (▼) | ΔLECE (▼) |
| Uncal | $2.12 \pm 0.26$ | $27.39 \pm 2.20$ | ✗ | $2.93 \pm 0.26$ | $10.72 \pm 0.68$ | ✗ | $4.63 \pm 0.07$ | $41.41 \pm 0.30$ | ✗ |
| RC | $1.09 \pm 0.07$ | $10.70 \pm 1.42$ | $4.73 \pm 1.55$ | $2.24 \pm 0.11$ | $10.50 \pm 1.10$ | $4.20 \pm 2.53$ | $3.73 \pm 0.07$ | $13.80 \pm 0.48$ | $8.66 \pm 2.98$ |
| T-CIL | $1.09 \pm 0.06$ | $10.27 \pm 1.01$ | $5.33 \pm 0.81$ | $2.23 \pm 0.10$ | $9.14 \pm 1.93$ | $10.05 \pm 7.17$ | $3.74 \pm 0.08$ | $\mathbf{10.23} \pm 1.34$ | $19.52 \pm 1.53$ |
| Ours | $\mathbf{1.08} \pm 0.06$ | $\mathbf{8.38} \pm 1.51$ | $\mathbf{-0.78} \pm 0.24$ | $\mathbf{2.21} \pm 0.11$ | $\mathbf{8.38} \pm 0.64$ | $\mathbf{-1.07} \pm 1.16$ | $\mathbf{3.71} \pm 0.07$ | $12.18 \pm 0.26$ | $\mathbf{-4.17} \pm 1.66$ |

**Test batch size**  In Table 20, we analyse the effect of the test batch size on the calibration performance. We can observe that changing the test batch size only marginally deteriorates the calibration error, suggesting that our approach can also work when partial information about the test set is available. In the edge case where only a single sample is available, it is possible to directly use the associated distance score as input for temperature selection.

Table 19: Effect of varying the test batch size on the calibration performance.

| | CIFAR100 | | | | | |
|---|---|---|---|---|---|---|
| Batch size | DATS | | | $\text{DATS}_w$ | | |
| | NLL (▼) | AECE (▼) | ΔLECE (▼) | NLL (▼) | AECE (▼) | ΔLECE (▼) |
| Full | $2.05 \pm 0.02$ | $7.92 \pm 0.56$ | $-0.82 \pm 1.66$ | $2.04 \pm 0.02$ | $7.88 \pm 0.60$ | $-3.13 \pm 0.23$ |
| 128 | $2.10 \pm 0.02$ | $8.12 \pm 0.36$ | $-2.08 \pm 1.39$ | $2.08 \pm 0.02$ | $8.27 \pm 0.67$ | $-2.85 \pm 0.51$ |
| 64 | $2.11 \pm 0.04$ | $8.57 \pm 0.60$ | $-1.59 \pm 0.39$ | $2.09 \pm 0.03$ | $8.48 \pm 0.64$ | $-2.72 \pm 0.60$ |
| 32 | $2.12 \pm 0.04$ | $8.56 \pm 0.64$ | $-2.36 \pm 0.29$ | $2.10 \pm 0.03$ | $8.48 \pm 0.68$ | $-2.77 \pm 0.70$ |
| 16 | $2.12 \pm 0.04$ | $8.65 \pm 0.76$ | $-2.23 \pm 0.42$ | $2.10 \pm 0.03$ | $8.64 \pm 0.56$ | $-2.51 \pm 0.80$ |
| 8 | $2.12 \pm 0.04$ | $8.70 \pm 0.76$ | $-1.81 \pm 0.99$ | $2.11 \pm 0.04$ | $8.79 \pm 0.64$ | $-2.86 \pm 0.29$ |
| 1 | $2.24 \pm 0.11$ | $9.84 \pm 0.79$ | $-0.48 \pm 0.76$ | $2.24 \pm 0.11$ | $9.84 \pm 0.79$ | $-0.48 \pm 0.76$ |

**Distance criterion**  In Table 20, we ablate the choice of the distance definition in order to test its effect on the calibration performance of our approach. From the results, it is clear that the minimum distance is the one delivering the better calibration results – this is in line with our intended goal (i.e., having a proxy of the closeness of each encountered class contained in $\mathcal{B}$ to the current ones).

Table 20: Effect of changing the distance criterion on the calibration performance.

| | CIFAR100 | | |
|---|---|---|---|
| Distance | DATS | | |
| | NLL (▼) | AECE (▼) | ΔLECE (▼) |
| Min | $2.05 \pm 0.02$ | $7.92 \pm 0.56$ | $-0.82 \pm 1.66$ |
| Max | $2.05 \pm 0.02$ | $8.56 \pm 0.58$ | $5.15 \pm 0.85$ |
| Avg | $2.32 \pm 0.04$ | $14.34 \pm 5.79$ | $1.05 \pm 3.55$ |

**Distance-Temperature Relationship Analysis**  To validate the theoretical foundation of DATS, we analyse the learned distance-temperature relationship on CIFAR100. After training, via optimisation on the calibration buffer, we learn the formula $T_t = 1.2335 \cdot d_t + 1.1362$ (Equation 7). Table 21 shows the computed distances $d_k$ and corresponding temperatures $T_k$ for each task $k \in \{1, \ldots, 10\}$ (using Equation 8) on the respective test sets. Consistent with our expectations, when $k = 10$ (the current task), the distance is near zero ($d_{10} = 0.0029$), yielding a temperature close to the base value ($T_{10} = 1.1397 \approx T_{\text{base}}$). For previous tasks, higher distances correspond to higher temperatures (Pearson's $r = 1.0$), confirming that miscalibration increases for samples outside recently trained

regions (Ovadia et al., 2019; Minderer et al., 2021). Furthermore, to verify that improvements stem from distance information rather than merely adding a learnable parameter, we conduct an ablation replacing actual distances with fixed constants during optimisation. Table 22 demonstrates that using complete distance information substantially outperforms the fixed variant across all datasets, confirming that distance scores carry intrinsic information essential for effective calibration.

Table 21: Distance-temperature relationship for each task on CIFAR100.

| Task ID ($k$) | $d_k$ | $T_k$ |
|---|---|---|
| 1 | 0.6098 | 1.8884 |
| 2 | 0.6544 | 1.9434 |
| 3 | 0.5633 | 1.8310 |
| 4 | 0.6364 | 1.9211 |
| 5 | 0.5733 | 1.8433 |
| 6 | 0.6371 | 1.9220 |
| 7 | 0.4602 | 1.7039 |
| 8 | 0.5111 | 1.7667 |
| 9 | 0.4496 | 1.6907 |
| 10 | 0.0029 | 1.1397 |

Table 22: Ablation study comparing complete distance information versus fixed distance values during optimization.

| Dataset | Complete | Fixed |
|---|---|---|
| | ECE (▼) | |
| CIFAR10 | $\mathbf{8.80} \pm 0.93$ | $11.07 \pm 3.11$ |
| CIFAR100 | $\mathbf{7.63} \pm 0.84$ | $9.61 \pm 0.30$ |
| TinyImageNet | $\mathbf{11.84} \pm 0.58$ | $13.72 \pm 0.33$ |
| Blood Cell | $\mathbf{9.87} \pm 1.40$ | $12.11 \pm 3.16$ |
| Skin Lesions | $\mathbf{12.53} \pm 2.68$ | $14.01 \pm 0.37$ |

## A.9 Pseudocode of DATS

---

**Algorithm 1** DATS training procedure.

---

1: **Input** $f_\theta$: trained model, $D_{t,val}$: validation set of task $t$, $\mathcal{B}_t$: calibration buffer.
2: **Notation** $t$: current task, $T$: number of tasks; $C_t$: set of classes in the validation set of current task; $\mathcal{C}_{buf}$: set of classes in the calibration buffer, $S$: distance matrix.
3:
4: **Class Distance Score Assignment**
5: **for** $c_{buf} \in \mathcal{C}_{buf}$ **do**                          ▷ For each class in $\mathcal{B}_t$
6:    $\boldsymbol{\mu}_{c_{buf}} \leftarrow$ GetClassPrototype$(f_\theta, c_{buf})$         ▷ Compute class prototype with Eq. 4
7: **end for**
8: **for** $c_t \in C_t$ **do**                          ▷ For each class in $C_t$
9:    $\boldsymbol{\mu}_{c_t} \leftarrow$ GetClassPrototype$(f_\theta, c_t)$         ▷ Compute class prototype with Eq. 4
10: **end for**
11: **for** $c_t \in C_t$ **do**
12:    **for** $c_{buf} \in \mathcal{C}_{buf}$ **do**
13:       $S[c_t, c_{buf}] \leftarrow$ Distance$(\boldsymbol{\mu}_{c_t}, \boldsymbol{\mu}_{c_{buf}})$         ▷ Compute pairwise distance with Eq. 5
14:    **end for**
15: **end for**
16: **for** $c_{buf} \in \mathcal{C}_{buf}$ **do**                          ▷ For each class in $\mathcal{B}_t$
17:    $d_{c_{buf}} \leftarrow$ AssignScore$(c_{buf})$         ▷ Assign class distance score with Eq. 6
18: **end for**
19:
20: **Distance-Aware Calibration**
21: $T(d_c) \leftarrow$ LearnTemperature$(d_c, \boldsymbol{z}_c)$         ▷ Compute class-wise temperatures using Eq. 7
22:
23: **Test-Time Calibration**
24: $\hat{C}_{\text{test}} \leftarrow$ AssignNearestClasses$(D_{t,test}, \{\boldsymbol{\mu}_{c_{buf}}\})$    ▷ Assign test sample to nearest $\boldsymbol{\mu}_{c_{buf}}$
25: $\hat{C}_{\text{test}} \leftarrow$ KeepFrequentClasses$(\hat{C}_{\text{test}}, 0.6)$    ▷ Keep frequent classes with $\geq 60\%$ coverage
26: $d_{\text{test}} \leftarrow$ ComputeTestDistance$(\hat{C}_{\text{test}}, \{d_c\})$         ▷ Compute test distance with Eq. 8
27: ApplyCalibration$(D_{t,test}, d_{\text{test}})$         ▷ Calibrate test logits based on $d_{\text{test}}$

---

