# OpenReview forum: "DATS: Distance-Aware Temperature Scaling for Calibrated Class-Incremental Learning"
_ICLR.cc/2026/Conference — Submitted to ICLR 2026_

### Official Review · Reviewer_DxNp · 2025-10-25

**Soundness:** 2
**Presentation:** 3
**Contribution:** 2
**Rating:** 4
**Confidence:** 4

**Summary:**

This paper introduces Distance-Aware Temperature Scaling (DATS), a post-hoc calibration method for Continual Learning (CL).
Instead of using a single global temperature for all tasks, DATS estimates the prototype-based distance between classes to infer task proximity and assign adaptive temperatures without task labels. By doing so, it aims to achieve more stable and reliable uncertainty calibration across sequential tasks. Experiments on standard benchmarks and biomedical datasets show that DATS effectively reduces calibration errors compared to existing calibration methods.

**Strengths:**

The paper is well written with clear logic and fluent expression, making it easy to read and understand. The motivation aligns well with the proposed method, and the presentation is visually appealing with informative figures and tables. The experiments are diverse and comprehensive, covering both benchmark and real-world datasets, and the results show consistently strong performance and clear advantages over existing methods.

**Weaknesses:**

The core assumption that class-prototype distances capture inter-task differences lacks theoretical support. The distance definition is heuristic and may be sensitive to noise. The experiments do not analyze the distance–temperature relationship, leaving the true contribution of distance unclear. The method is only tested on the ER baseline, limiting claims of generality, and no analysis is provided on the effect of memory buffer size.

**Questions:**

1. The core assumption of this paper is that inter-task distributional differences can be captured through class-prototype distances. However, in practice, class prototypes only reflect differences in the mean of the distributions, while variations at the level of distributional variance cannot be represented. Therefore, the motivation lacks theoretical justification and cannot be considered fully valid.
2. The distance definition in Equation (6) is somewhat heuristic, as using the minimum distance makes it overly sensitive to noisy or outlier classes. It is recommended to include an additional experiment comparing results obtained by replacing the minimum distance with average, Top-K, or weighted similarity measures.
3. The experimental section lacks an analysis of the distance–temperature relationship, since Equation (7) essentially adds a per-class temperature bias. The experiments do not examine how ($ d_c $) actually relates to the learned temperature, and the observed improvements might simply result from the parameter ($ \omega_c$ ). What would happen if ( $d_c$ ) were fixed rather than representing a distance? It is recommended to include such comparisons to address potential doubts.
4. The proposed method appears to be a plug-in approach with claimed general applicability, and it is evaluated on top of Experience Replay (ER). However, ER is a relatively classical CIL baseline with considerable room for improvement. If the proposed method were deployed on more advanced CIL frameworks, would it still demonstrate such a level of effectiveness?
5. Although the paper employs a memory buffer, the experiments do not include any analysis of the buffer size. It is recommended to add such experiments to assess the method’s sensitivity to different memory capacities.

---

> ### Author Response · Authors · 2025-11-21
> **Response to Reviewer DxNp**
>
> Thank you *Reviewer DxNp* for the detailed and helpful comments. Below are our responses to your questions.
>
> 1. **Motivation lacks theoretical justification**:
> The motivation of our approach stems from the empirical observations summarised in Figures 3 and 4. As explained in response to *Reviewer xTqV*, the results reported in Figure 3 corroborate the hypothesis that the classifier is biased toward the last task since we can observe how the average confidence (and the accuracy) of the model for the current task is higher than for older tasks. However, for older tasks, we believe that the confidence does not decrease at the same rate as the accuracy does (due to catastrophic forgetting). For this reason, we hypothesise that older tasks have a higher discrepancy between accuracy and confidence (leading to higher calibration error) that should be fixed with a higher temperature. An easy and effective way of checking whether a set of samples is from the current task or not is by measuring its distance from class prototypes and use this measure as an additional signal to tune the temperature accordingly. We have also explored sample-wise approaches via sample-specific temperatures to account for the distributional variance you mention (see the results with batch size = 1 in response to **Assumption on batched test data** for *Reviewer pqgq*) but found empirically that this did not improve performance.
>
> 2. **Distance definition is heuristic**:
> The distance defined in Equation 6 is justified by our hypothesis explained above. For each encountered class (i.e., the classes in the calibration buffer), we want to have an idea of its closeness to the current ones. If the distance is very small (i.e., approximately 0), we can assume the given class to be part of the current ones, which need less calibration than those with higher distance. This motivates us to use the minimum distance between class prototypes as a proxy of the closeness between classes. In order to test the effect of this choice, we have followed your suggestion and ablate the choice of the distance definition. From the results, it is clear that the minimum distance is the one delivering the better calibration results. We be believe this is the case since it aligns with our intended goal.
>
> | Distance | NLL (↓) | AECE (↓) | ΔLECE (↓) |
> |----------|---------|-----------|-----------|
> | Min | 2.05 ± 0.02 | 7.92 ± 0.56 | -0.82 ± 1.66 |
> | Max | 2.05 ± 0.02 | 8.56 ± 0.58 | 5.15 ± 0.85 |
> | Avg | 2.32 ± 0.04 | 14.34 ± 5.79 | 1.05 ± 3.55 |
>
>
> 3. **Distance-temperature relationship analysis**
> We analyse the distance–temperature relationship following your suggestion in the table below. Given a trained model on CIFAR100, via the calibration buffer we learn the following formula (Equation 7) to tune the temperature based on the distance: $T_t = 1.2335 * d_t + 1.1362$. Afterwards, for each task $k \in {1, \dots, t}$, we compute the distance $d_k$ using the corresponding test set via Equation 8.
> Below, we report the computed distance $d_k$ and the corresponding temperature $T_k$ for each considered task. In line with our expectations, when the task ID matches the current one (i.e., $k=t$), the computed distance is approximatively 0. This leads to a reduced applied temperature (basically just $T_{base}$ in Equation 7).
>
> | Task ID (k) | d_k | T_k |
> | - | - | - |
> | 1          | 0.6098 | 1.8884 |
> | 2          | 0.6544 | 1.9434 |
> | 3          | 0.5633 | 1.8310 |
> | 4          | 0.6364 | 1.9211 |
> | 5          | 0.5733 | 1.8433 |
> | 6          | 0.6371 | 1.9220 |
> | 7          | 0.4602 | 1.7039 |
> | 8          | 0.5111 | 1.7667 |
> | 9          | 0.4496 | 1.6907 |
> | 10         | 0.0029 | 1.1397 |
>
> 4. **Limited integration with CIL algorithms**:
> To demonstrate the verstility and plug-and-play character of our approach we followed your suggestion (and those of *Reviewer pqgq* and *Reviewer xTqV*), and report new results on additional CIL algorithms (EWC, LwF, RAR, and GDumb). The results demonstrate that our approach can indeed be applied in a plug-and-play manner to diverse CL strategies and consistently reduces ECE in all cases.
>
> 5. **Memory size ablation study**:
> For simplicity, we set the size of the calibration buffer equal to the size of the memory buffer used during training. Changing the memory buffer would only change the performance of the backbone method. What happens when the performance of the backbone model changes has been implicitly evaluated with our additional experiments, where the performance of the backbone model varies depending on the employed CIL strategy. These experiments show that DATS also effectively reduces calibration error in the low-accuracy regime.  In addition, in Appendix A.5 - Table 7, we already the size of the calibration buffer (which directly affects our calibration procedure) for different datasets. The results suggest that the size of the calibration buffer has only a minor effect on the calibration performance of our approach.

---

> ### Comment · Reviewer_DxNp · 2025-11-23
>
> We sincerely thank the authors for their detailed responses and for providing additional experimental results. We acknowledge the effort invested in addressing our concerns. Below we summarize the remaining issues and suggestions.
>
> 1. On the use of prototypes as distributional representatives
> While we appreciate the authors’ explanations, the response still does not clarify why class prototypes—being only the means of the feature distributions—can sufficiently characterize inter-task distributional differences. The current justification remains empirical and lacks theoretical grounding. Since this assumption is tightly linked to the motivation of the paper, we encourage the authors to provide a more rigorous explanation or supporting evidence. A convincing justification here would significantly influence our final rating.
>
> 2. On the distance definition
> The clarification regarding the distance definition is reasonable, and we agree that this part no longer requires further discussion. However, this explanation implicitly relies on the assumption that prototypes are valid proxies for class distributions, which is precisely the concern raised in Point 1. We therefore encourage the authors to address the prototype–distribution relationship more thoroughly.
>
> 3. On distance–temperature relationship analysis
> The current response does not provide experimental or theoretical evidence demonstrating the necessity of mapping distances to temperatures. Given that Equation (7) essentially introduces a per-class temperature bias, it remains unclear whether the improvements stem from the distance-based design or simply from an additional learned parameter. We encourage the authors to provide an analysis of how distances correlate with learned temperatures, or a comparison where distances are replaced with fixed constants.
> Such results would significantly strengthen the technical soundness of the proposed mechanism.
>
> 4. On the plug-and-play property
> The additional experiments demonstrating the plug-and-play capability are sufficiently convincing. We have no further concerns regarding this aspect.
>
> 5. On the calibration buffer size
> In the rebuttal, the authors stated:
> “in Appendix A.5 – Table 7, we already the size of the calibration buffer …”
> This sentence appears incomplete, and we were unable to locate any buffer-size analysis in Appendix A.5 or Table 7. We kindly request the authors to supplement this missing analysis or correct the reference if it was mislabelled.

---

> ### Author Response · Authors · 2025-11-27
> **Response to Official Comment by Reviewer DxNp (1/2)**
>
> Thanks *Reviewer DxNp* for your prompt feedback. Below is our response to the remaining concerns.
>
> 1 & 2. We agree that class prototypes do not fully characterise the underlying feature distribution and cannot capture higher-order statistics. However, our method does not require full distributional modelling as we outline below. We now provide the theoretical assumptions under which prototype-based distances are justified for our approach, along with their connection to temperature scaling and task recency.
>
> - (a) *Class means are theoretically sufficient under standard feature-space assumptions.*
> Many analyses of deep representation spaces (e.g., Mahalanobis-based OOD detectors [1]) assume that penultimate-layer features are locally Gaussian with *approximately shared within-class covariance* across tasks:
> $$
> h \mid (y=c,t) \sim \mathcal{N}(\mu_{t,c}, \Sigma),
> \quad \Sigma_{t,c} \approx \Sigma.
> $$
> Under this shared-covariance model, all standard distributional discrepancies between two class-conditional distributions (e.g., KL divergence, Mahalanobis distance, and Wasserstein-2) reduce to functions of the mean difference:
> $$ \mathrm{KL}\left(\mathcal{N}(\mu_{t,c},\Sigma)\,\|\,\mathcal{N}(\mu_{t',c},\Sigma)\right) = \tfrac{1}{2}(\mu_{t,c}-\mu_{t',c})^\top \Sigma^{-1}(\mu_{t,c}-\mu_{t',c}).
> $$
> Thus, the class mean is a sufficient statistic for quantifying inter-task drift at the class level in this setting. Our prototype is simply the empirical estimator of this mean, and our cosine-based distance serves as a practical surrogate for such Mahalanobis-style divergences. Class-incremental learning can be viewed as a form of temporal distribution shift, where old task classes become "out-of-distribution" relative to the most recent training regime. Even when the Gaussian shared-covariance assumption does not hold exactly, using penultimate layer features has been demonstrated to work well in OOD scenarios since the penultimate layer preserves relevant information for the task [2].
>
>
> - (b) *Alignment with capacity of temperature scaling.*
> Temperature scaling adjusts only a single scalar. It can correct global logit-scale mismatches but cannot account for differences in covariance or higher-order statistics. Therefore, using first-order statistics (i.e., class means) is not a limitation but matches exactly the expressive capacity of temperature scaling. Incorporating higher-order statistics would provide information that TS cannot exploit. Furthermore, the corresponding estimates would be highly unstable given the limited-size calibration buffer. The calibration buffer $\mathcal{B}_t$ typically contains very few samples per class (e.g., $N=20$) in a high-dimensional feature space (e.g., $D=512$).
>
>     - For the estimation of the mean ($\mu$), the error rate converges at $O(\sqrt{D/N})$.
>
>     - For the estimation of the covariance ($\Sigma$), the error rate is significantly higher requiring $O(D^2)$ parameters. With $N \ll D$, the sample covariance matrix becomes singular and ill-conditioned making reliable estimation infeasible.
>
> To summarise, while prototypes do not encode the entire class distribution, they provide exactly (i) the theoretically justified first-order statistics that dominate inter-task drift under shared-covariance models, and (ii) the level of information that temperature scaling can meaningfully exploit. This makes prototype distances an appropriate and principled choice for our calibration framework. Furthermore, the effectiveness of our method across multiple architectures, CL baselines, and datasets provides strong empirical evidence that prototype distances are the primary driver of calibration errors in this setting.

---

> > ### Author Response · Authors · 2025-11-27
> > **Response to Official Comment by Reviewer DxNp (2/2)**
> >
> > 3. We would like to kindly point out that we do provide an empirical analysis of the distance-temperature correlation in the table reported in response to **Distance-temperature relationship analysis**. From the results, it is clear that, when $d_k \simeq 0$, the temperature is (almost) reduced to $T_{base}$. Regarding the other tasks, the higher is the distance, the higher is the temperature – confirming that there is a perfect correlation (Pearson's r) between distance and miscalibration. This is consistent with prior analyses on calibration under distribution shift [3,4], where confidence errors increase when inputs fall outside recently trained regions.
> > Regarding the experiment with fixed distance at test time, we would like to politely highlight that this is basically equivalent to the experiments on Replayed Calibration (RC) – which we demonstrate to be worse than our approach. In fact, learning a single temperature is equivalent to assuming all samples in the test set have the same distance.
> > To assess on whether the introduction of an additional learned parameter is the main reason for the improvement, we now report below a new experiment where the distance information provided during the optimisation phase is replaced by a constant number. From the results, it is clear that the additional learned parameter is not the responsible factor for the improved calibration. Instead, this further demonstrates that the distance scores carry intrinsic information that is practically relevant for calibration.
> >
> > | Distance information | Complete | Fixed |
> > |:-:|:-:|:-:|
> > |  | **ECE (↓)** | **ECE (↓)** |
> > | CIFAR10 | 8.80 ± 0.93 | 11.07 ± 3.11 |
> > | CIFAR100 | 7.63 ± 0.84 | 9.61 ± 0.30 |
> > | TinyImageNet | 11.84 ± 0.58 | 13.72 ± 0.33 |
> > | Blood Cell | 9.87 ± 1.40 | 12.11 ± 3.16 |
> > | Skin Lesions | 12.53 ± 2.68  | 14.01 ± 0.37 |
> >
> > 4. We are glad to know that this point has been addressed.
> >
> > 5. We are sorry for the confusion. After the revision of the paper, the table with the ablation study of the calibration buffer size is now in Appendix A.7 - Table 17. The table describes the effect on the calibraton performance when we change the percentage of retained samples from the validation set of each task. From the results (now also reported below), we can see that changing the size of the calibration buffer (reported as the percentage of retained samples from each validation set $D_{t, val}$) has only a minor impact on the calibration performance of our approach.
> >
> > | Val. inclusion (%) | 20 | 30 | 40 | 50 |
> > |:-:|:-:|:-:|:-:|:-:|
> > |  | **ECE (↓)** | **ECE (↓)** | **ECE (↓)** | **ECE (↓)** |
> > | CIFAR10 | 8.85 ± 0.81 | 8.99 ± 0.68 | 9.02 ± 0.67 | 8.83 ± 0.75 |
> > | CIFAR100 | 7.50 ± 0.55 | 7.44 ± 0.95 | 7.63 ± 0.84 | 7.81 ± 0.67 |
> > | TinyImageNet | 11.84 ± 0.58 | 11.56 ± 0.85 | 11.67 ± 0.81 | 11.46 ± 0.82 |
> >
> >
> > [1] Lee, Kimin, et al. "A simple unified framework for detecting out-of-distribution samples and adversarial attacks." Advances in neural information processing systems 31 (2018).
> >
> > [2] Sun, Yiyou, et al. "Out-of-distribution detection with deep nearest neighbors." International conference on machine learning. PMLR, 2022.
> >
> > [3] Ovadia, Yaniv, et al. "Can you trust your model's uncertainty? evaluating predictive uncertainty under dataset shift." Advances in neural information processing systems 32 (2019).
> >
> > [4] Minderer, Matthias, et al. "Revisiting the calibration of modern neural networks." Advances in neural information processing systems 34 (2021): 15682-15694.

---

### Official Review · Reviewer_y6Qa · 2025-11-01

**Soundness:** 2
**Presentation:** 3
**Contribution:** 2
**Rating:** 6
**Confidence:** 3

**Summary:**

This paper concentrates on the issue of calibration in continual learning and propose a temperature based method, i.e., Distance-Aware Temperature Scaling (DATS), which combines prototype-based distance estimation with distance-aware calibration to infer task proximity and assign adaptive temperatures without prior task information. The main contributions are summarized as:

a. This paper demonstrates that calibration should move from task-agnostic to task-aware and adaptive re-calibration methods and expose the limitations of current task-agnostic data-centric approaches.

b. This paper proposes a distance-aware calibration framework that shifts the focus from data selection and injects task-awareness directly into the re-calibration algorithm.

c. Provide extensive experiments to prove the effectiveness of the proposed method.

**Strengths:**

a. This paper is well written and easy to follow.

b. It is interesting to consider the issue of calibration in continual learning.

c. This paper is technically solid.

**Weaknesses:**

a. My major concern is about the theoretical analysis. In Section 2, this paper introduces the definition of perfect calibration. I would like to know how the calibration results obtained by the proposed method compare, in theory, to those of a perfectly calibrated model. Please provide a detailed theoretical analysis.

b. Please compare this paper with [1] in details.

[1] Calibration of continual learning models. In CVPR 2024.

**Questions:**

N/A

---

> ### Author Response · Authors · 2025-11-21
> **Response to Reviewer y6Qa**
>
> Thank you *Reviewer y6Qa* for the positive and valuable feedback. Below are our responses to your questions.
>
> 1. **Implications of perfect calibration**:
> For a more throrough analysis of perfect  calibration, we build on recent theoretical work on proper calibration errors (Gruber & Buettner, NeurIPS 2022). This framework has two implications for our work. First, Theorem 3.1 shows that perfect calibration implies $ECE=0$. That means, if we assume perfect calibration as you suggest, the expected calibration error of the model under consideration will provably be 0. Second, from Proposition 4.5 it follows that – since TS-based DATS is an injective transformation of the logits – we can quantify the canonical calibration improvement of DATS via a simple upper bound; assuming the L2 calibration error, this can be estimated via the square root of the Brier score. If again we assume perfect calibration, this result tells us that DATS would not change the Brier score of the inital model.
>
> Gruber, S., & Buettner, F. (2022). Better uncertainty calibration via proper scores for classification and beyond. Advances in Neural Information Processing Systems, 35, 8618-8632.
>
> 2. **Comparison with [1]**:
> We do provide a comparison with [1] in the paper. As detailed in Section 2 of our manuscript, [1] represents the first study investigating the problem of calibration in CIL settings. The authors propose to employ a calibration buffer filled with samples taken from the validation sets to learn a single temperature for all tasks. The authors focus on the choice of the data to use (i.e., random selection of validation samples) but overlook intra-task variability leading to less effective calibration than our approach. We show that DATS consistently outperfroms this baseline.
>
> [1] Li, Lanpei, et al. "Calibration of continual learning models." Proceedings of the IEEE/CVF Conference on Computer Vision and Pattern Recognition. 2024.

---

### Official Review · Reviewer_xTqV · 2025-11-01

**Soundness:** 3
**Presentation:** 3
**Contribution:** 2
**Rating:** 4
**Confidence:** 1

**Summary:**

This paper introduces the concept of task-aware calibration into the field of continuous learning and proposes the DATS method. It employs category prototype distance as a proxy for task dissimilarity to achieve adaptive calibration without task labels. The article first reveals the limitations of existing approaches through robust empirical analysis and innovatively introduces two new metrics, ΔLECE and max ΔECE, providing fresh perspectives for evaluating calibration stability. The core of the approach lies in inferring task divergence through prototype distances, alongside designing a lightweight runtime inference mechanism. However, the method lacks validation of buffer prototype stability. Furthermore, the paper fails to adequately explain why greater task distances necessitate higher temperatures, and its fixed-threshold runtime calibration mechanism may falter when model accuracy is low. Its validity is primarily validated within the Experiential Replay (ER) framework, lacking experimental support across other continuous learning paradigms. Overall, this paper makes significant contributions in problem definition, methodological innovation, and evaluation systems. However, the reliability of its conclusions could be further strengthened through deeper validation of core assumptions and broader experimental testing.

**Strengths:**

1. The paper precisely identifies and substantiates a critical issue. Through experiments (as shown in Figures 3 and 4), it reveals that existing ‘task-agnostic’ approaches cause model confidence to drift between new and old tasks, providing empirical grounds for the necessity of ‘task-aware’ calibration.

2. The authors propose the DATS method, which ingeniously addresses core constraints of task-sensitivity in real-world deployment. It quantifies task differences through ‘category prototype distance’ and designs a lightweight test-time inference mechanism.

3. The paper innovatively introduces two metrics: ΔLECE (impact on the latest task) and max ΔECE (worst-case performance degradation), precisely quantifying calibration stability and reliability.

4. Validation on biomedical datasets exhibiting category imbalance robustly demonstrates the method's resilience and practical value in complex data environments.

**Weaknesses:**

1. The core of the method lies in the cosine distance of class prototypes at the penultimate layer. Due to catastrophic forgetting, feature representations of older classes may degrade, and the stability of prototypes in the calibration buffer has not been verified.

2. At low accuracy levels, the fixed-threshold calibration mechanism may misclassify the current task as another task, severely compromising calibration effectiveness.

3. The paper assigns higher temperatures to tasks farther from the current task, yet fails to explain why greater distance warrants increased temperature.

4. The paper primarily employs Experiential Replay (ER) as the continuous learning training method, lacking experiments on other types of CL algorithms.

**Questions:**

1. The author should explain how to ensure the stability and reliability of older class prototypes in the calibration buffer, given that feature representations are susceptible to catastrophic forgetting.

2. The author should give a detailed explanation about the fixed-threshold calibration mechanism prevent misclassifying the current task when its accuracy is low and its prototype is close to a different task's. A sensitivity analysis of the threshold may be needed.

3. What is the justification for assigning a higher temperature to tasks that are farther away? The author is suggested to provide theoretical or empirical explanation for this design choice.

4. The experiments are primarily based on Experience Replay (ER). Can the demonstrated effectiveness of the calibration method be generalized to other families of continual learning algorithms?

---

> ### Author Response · Authors · 2025-11-21
> **Response to Reviewer xTqV**
>
> Thank you *Reviewer xTqV* for your thorough and helpful comments. Below are our responses to your concerns.
>
> 1. **Stability and reliability of older class prototypes**:
> It is true that the class prototypes extracted from the penultimate layer may deteriorate due to catastrophic forgetting. However, thanks to the quality of the representations of the last task, we preserve discriminative power between the last task and the previous ones. That is, we expect the distance between prototype classes to be almost 0 for those belonging to the last task and $>0$ otherwise (we verified this in our response to **Distance-temperature relationship analysis** for *Reviewer DxNp*, where we provide a numerical example of the computed distance scores per task). This is preserved also in low-accuracy regimes as demonstrated in the additional experiments with weaker baselines that we provide (e.g., LwF and GDumb on TinyImageNet).
>
>
> 2. **Limitations of fixed-threshold mechanism**:
>  An analysis of the threshold sensitivy is already present in Appendix A.5 - Table 6. From the results, we see that reducing the threshold usually improves the final ECE since we filter out the least present classes and, most probably, the misclassified ones. The results on additional CIL algorithms suggest that, in the case the baseline method performs poorly, our approach is still able to reduce ECE considerably anyway. Furthermore, we have now also evaluated a threshold-free approach to not depend on this hyperparameter. To this end, simply take the weighted average of the distance scores. In other words, we can replace Equation 8 with:
> $$ d_{\text{test}} = \sum_{c \in \hat{C}}  \frac{n_c}{|D_{\mathrm{t,test}}|} d(c_{\mathrm{buf}}=c) $$
>  where $\hat{C}$ represents the set of assigned classes and $n_c$ represents the number of samples in $D_{\text{t,test}}$ assigned to class $c \in \hat{C}$. We denote the weighted strategy as $DATS_w$. The results of $DATS_w$ are comparable to $DATS$ on all the considered datasets and strategies, demonstrating its effectiveness while avoiding tuning the threshold explicitly.
>
> 3. **Justification for higher temperatures to farther tasks**:
>  As highlighted by *Reviewer pqgq* and *Reviewer D1Dg*, the classifier may be biased toward the last task. The results reported in Figure 3 corroborate this hypothesis since we can observe how the confidence of the model decreases for previous tasks. However, for older tasks, our intuition is that the confidence does not decrease at the same pace as the accuracy does (due to catastrophic forgetting). For this reason, we hypothesise that farther tasks have a higher difference between accuracy and confidence that should be fixed with a higher temperature.
>
> 4. **Limited integration with CIL algorithms**:
> To demonstrate the verstility of our approach we followed your suggestion (and those of *Reviewer pqgq* and *Reviewer DxNp*), and report new results on additional CIL algorithms (EWC, LwF, RAR, and GDumb). The results demonstrate that our approach is not sensitive to the choice of the employed CL strategy and consistently reduces ECE in all cases.

---

> > ### Comment · Reviewer_xTqV · 2025-11-27
> >
> > I have read the response carefully, the authors have addressed all my questions, I have no other concerns. And I will keep the score unchanged.

---

### Official Review · Reviewer_D1Dg · 2025-11-01

**Soundness:** 2
**Presentation:** 3
**Contribution:** 2
**Rating:** 2
**Confidence:** 4

**Summary:**

This paper introduces Distance-Aware Temperature Scaling (DATS), a method designed to improve model calibration in continual learning settings. The approach combines prototype-based distance estimation with distance-aware calibration to infer task proximity and assign adaptive temperatures that shape predictive distributions accordingly. The authors conduct empirical evaluations across both standard continual learning benchmarks and real-world datasets from the biomedical domain, demonstrating that their approach consistently reduces calibration error across tasks compared to baseline approaches.

**Strengths:**

- The proposed method is straightforward to implement and demonstrates clear effectiveness in improving calibration performance.
- The empirical evaluation is comprehensive, incorporating both standard continual learning benchmarks and realistic datasets from the biomedical domain, which strengthens the practical relevance of the findings.
- The paper is generally easy to follow.

**Weaknesses:**

- Tables 1 and 2 lack direct comparisons of classification performance metrics (e.g., accuracy) between the proposed method and baseline approaches. While DATS shows improved calibration, it is essential to verify that classification performance remains competitive with existing methods. Although NLL provides some indication of predictive quality, it does not directly substitute for standard accuracy metrics.

- The test-time calibration procedure described in Section 3.3 appears to require inference in a transductive setting, where optimal temperatures are determined collectively across a batch of test samples. This setup may inherently favor calibration performance. It would be valuable to demonstrate that the method maintains both improved calibration and competitive classification performance under a more challenging inductive setting, where test samples are processed individually rather than in batches.

My current rating is primarily influenced by these two weaknesses, I am willing to revise my score if they can be adequately addressed.

**Questions:**

- Poor calibration in continual learning often stems from continuously updating the final linear classifier, which can introduce bias toward recent tasks (as illustrated in Figure 3). How does the proposed method compare to simpler baselines that address this issue directly, such as GDumb [1] or other continual learning approaches that retrain the final classifier layer from scratch after each learning episode? Understanding this comparison would help clarify whether the calibration improvements are primarily due to the temperature scaling mechanism or could be achieved through alternative classifier alignment strategies.

---

> ### Author Response · Authors · 2025-11-21
> **Response to Reviewer D1Dg**
>
> Thank you *Reviewer D1Dg* for your detailed and helpful feedback. Below are our responses to your concerns.
>
> 1. **Lack of classification performance metrics**:
>  Being a post-hoc calibration method based on accuracy-preserving temperature scaling, the classification performance is not affected by our method. Indeed, this is considered one of the key properties any uncertainty calibration method should have (Zhang et al, ICML 2020). For this reason, in all the tables, the accuracy of the trained model is reported in parenthesis next to the name of the considered dataset.
>
> Zhang, J., Kailkhura, B., & Han, T. Y. J. (2020). Mix-n-match: Ensemble and compositional methods for uncertainty calibration in deep learning. ICML
>
> 2. **Transductive/inductive setting at test-time**:
> We agree that the inductive setting is indeed of relevance and now report the effect of varying the size of the test batch size on the calibration performance, including the size 1, where test samples are processed individually. From the result tables (see response to **Assumption on batched test data** for *Reviewer pqgq*), we can observe that changing the test batch size only marginally deteriorates the calibration error, suggesting that our approach can also work when partial information about the test set is available. In the edge case where only a single sample is available, we directly use the associated distance score as input for temperature selection.
>
> 3. **Comparison with GDumb**:
> We agree that it is of interest to further assess the well known bias towards recent tasks. Your comment is in line with *Reviewer pqgq*, who suggested to combine a baseline that explicitly addresses this bias with post-hoc calibration. We therefore ran additional experiments with GDumb, first as baseline as is and second in combination with DATS as post-hoc calibration strategy.  While we had to increase the memory size for each of the considered dataset to achieve competitive accuracy, GDumb is indeed able to deliver better calibration results: the average ECE (AECE) is lower than the one for other baselines such as ER. This is indeed probably due to the classifier being less biased toward the last task. Importantly though, when we applied our approach on top of GDUmb, we found that the calibration error could consistently be further reduced. This suggests that there is a benefit induced by our task-aware temperature scaling approach that goes beyond bias mitigation via stabelized representations.
>  These results also illustrate the plug-and-play nature of our method. DATS as  post-hoc recalibration strategy can be used in combination with a wide range of CIL strategies and does not require a specific training setup that may not be suitable for a given application; we show also for diverse other strategies that it consistently improves model calibration (see response to **Limited integration with CIL algorithms** for *Reviewer pqgq*).
>
>
>  **GDumb**
>
>  | Method | NLL (↓) CIFAR10 | AECE (↓) CIFAR10 | ΔLECE (↓) CIFAR10 | NLL (↓) CIFAR100 | AECE (↓) CIFAR100 | ΔLECE (↓) CIFAR100 | NLL (↓) TinyImageNet | AECE (↓) TinyImageNet | ΔLECE (↓) TinyImageNet |
> |-|:-:|:-:|:-:|:-:|:-:|:-:|:-:|:-:|:-:|
> | Acc | 33.35 ± 3.25 |  |  | 24.27 ± 0.57 |  |  | 10.59 ± 1.59 |  |  |
> | GDumb | 1.91 ± 0.06 | 14.71 ± 2.44 | &cross; | 3.42 ± 0.11 | 15.03 ± 2.75 | &cross; | 4.38 ± 0.08 | 5.29 ± 0.48 | &cross; |
> | +DATS | **1.83 ± 0.05** | 12.68 ± 2.04 | **1.61 ± 0.91** | **3.16 ± 0.03** | **3.79 ± 0.10** | **-7.56 ± 5.46** | **4.29 ± 0.10** | 2.96 ± 0.16 | -1.97 ± 3.39 |
> | +DATS_w | 1.83 ± 0.06 | **12.54 ± 2.17** | 3.42 ± 2.01 | **3.16 ± 0.03** | 3.81 ± 0.19 | -7.46 ± 5.39 | **4.29 ± 0.10** | **2.93 ± 0.16** | **-2.02 ± 3.34** |
>
> | Method | NLL (↓) Blood Cell | AECE (↓) Blood Cell | ΔLECE (↓) Blood Cell | NLL (↓) Skin Lesions | AECE (↓) Skin Lesions | ΔLECE (↓) Skin Lesions |
> |-|:-:|:-:|:-:|:-:|:-:|:-:|
> | Acc | 68.94 ± 13.57 |  |  | 46.95 ± 4.02 |  |  |
> | GDumb | 1.36 ± 0.76 | 19.65 ± 10.99 | &cross; | 1.50 ± 0.25 | 15.97 ± 6.70 | &cross; |
> | +DATS | **0.85 ± 0.36** | **8.85 ± 5.43** | **-1.10 ± 3.42** | 1.44 ± 0.10 | 14.97 ± 3.73 | 0.47 ± 9.30 |
> | +DATS_w | **0.85 ± 0.36** | 8.99 ± 5.02 | 4.85 ± 0.94 | **1.42 ± 0.13** | **13.36 ± 5.24** | **-0.47 ± 12.06** |

---

### Official Review · Reviewer_pqgq · 2025-11-01

**Soundness:** 3
**Presentation:** 3
**Contribution:** 2
**Rating:** 2
**Confidence:** 4

**Summary:**

This paper addresses the underexplored problem of uncertainty calibration in class-incremental continual learning (CL), where models must maintain both accuracy and reliable confidence estimates as new tasks arrive. Existing methods like RC and T-CIL use a single temperature across all tasks, ignoring task-specific variability caused by catastrophic forgetting, which leads to unstable calibration. The authors propose Distance-Aware Temperature Scaling (DATS), a post-hoc calibration framework that infers task proximity through prototype-based distance estimation and adjusts the temperature adaptively without requiring task labels at test time. By combining distance-aware calibration with prototype matching, DATS introduces task-awareness into temperature optimization. Extensive experiments on standard benchmarks (CIFAR-10/100, TinyImageNet) and imbalanced biomedical datasets (BloodCell, SkinLesions) show that DATS significantly reduces Expected Calibration Error (ECE) and maintains stability across tasks, outperforming task-agnostic baselines while remaining computationally efficient.

**Strengths:**

* Clear identification of prior limitations: The paper clearly diagnoses the limitations of existing continual learning (CL) calibration methods, which rely on a single, task-agnostic temperature, failing to account for inter-task variability caused by catastrophic forgetting.
* Improved calibration stability: The results demonstrate that DATS consistently reduces Expected Calibration Error (ECE) and avoids severe miscalibration across tasks—especially valuable in medical and safety-critical applications.
* Efficiency and practicality: Despite introducing distance-based components and task-tailored temperature scaling, DATS remains computationally efficient compared to T-CIL and simple to integrate as a post-hoc method.

**Weaknesses:**

* Misattributed source of miscalibration: The paper attributes inter-task miscalibration mainly to intrinsic differences in task uncertainty, but from the experiment results, the main source of the task uncertainty difference would be a well-known artifact of the class-incremental learning (CIL) setup itself rather than the task difference—specifically, the phenomenon that logits and confidence scores become biased toward the current task. Prior works (e.g., SS-IL, BiC, or bias-correction-based CIL methods) have shown that this stems from representation drift and imbalance in CIL learning, rather than from a true difference in task-level uncertainty. Hence, applying task-dependent temperature scaling might address this symptom. Hence, it would be better to first stabilize representation learning or balance logit distributions before introducing per-task calibration. Furthermore, from the experimental results in this paper, the previous tasks uncertainty gap look much smaller than the gap between previous and current tasks. This paper needs more persuasive experimental setup that can better exhibit the motivation of this paper.
* Assumption of batched test data. A major limitation is that DATS requires access to a set of test samples to compute representative class assignments and estimate the dominant class distances. This assumption does not hold in streaming or online inference scenarios, where test samples arrive one by one.
* Limited integration with CIL algorithms. The proposed method is evaluated as a post-hoc module, independent of the underlying CIL training dynamics. However, calibration behavior in CIL is tightly linked to how tasks are learned and replayed. Without jointly considering the training algorithm (e.g., ER, DER, or SS-IL variants), the effect of DATS may vary, making its generality uncertain.
* Novelty boundary. The novelty lies mainly in introducing a distance-aware temperature modulation rather than a fundamentally new calibration formulation. The method extends existing temperature scaling ideas with an additional distance-dependent weighting, which might be considered incremental.

**Questions:**

The proposed DATS framework assumes that a representative set of test samples is available to infer task proximity and compute adaptive temperatures. However, in many continual learning scenarios, test samples may arrive sequentially (e.g., in streaming or online inference). How would DATS handle such settings?

---

> ### Author Response · Authors · 2025-11-21
> **Response to Reviewer pqgq (1/3)**
>
> We thank *Reviewer pqgq* for the constructive and valuable comments. Below are our responses to your concerns and questions.
>
> 1. **Misattributed source of miscalibration**:
>  We agree that a source of miscalibration is the bias of logits and confidence scores toward the current task.  *Reviewer D1Dg* has raised a very similar question and suggested to use GDUmb as a baseline in order to evaluate the effect of directly correcting this bias. We therefore first introduced GDumb as baseline that addresses this issue directly and then use GDumb to re-calibrate these more stable representations as you suggest. The full results tables are reported in response to *Reviewer D1Dg*. What we find is that indeed GDumb is able to deliver better calibration results than other CL strategies: the average ECE (AECE) is lower than the one reported for other baselines, which is most likely due to the more balanced logit distribution resulting form the training procedure. Importantly though,  when we applied our approach on top of GDUmb, we found that the calibration error could consistently be further reduced. This suggests that there is a benefit induced by our task-aware temperature scaling approach that goes beyond stabilized representations. These results also illustrate the plug-and-play nature of our method. DATS as an accuracy-preserving post-hoc recalibration strategy can be used in combination with a wide range of CIL strategies and does not require a specific training setup that may not be suitable for a given application; we show also for diverse other strategies that it consistently improves model calibration (see response to **Limited integration with CIL algorithms**).
>
>
> 2. **Assumption on batched test data**:
> We agree that online inference scenarios where single test sample arrive one at a time is an important scenario. We therefore conducted additional experiments which show that DATS is indeed able to perfrom well across a wide range of test batch sizes, including single samples. From the tables below, we observe that changing the test batch size only marginally deteriorates the calibration, suggesting that our approach can also work when partial information about the test set is available.
>
> **Effect of the test batch size on the calibration performance when using $DATS$**
> | Batch size | NLL (↓) | AECE (↓) | ΔLECE (↓) |
> |:-|:-:|:-:|:-:|
> | Full | 2.05 ± 0.02 | 7.92 ± 0.56 | -0.82 ± 1.66 |
> | 128  | 2.10 ± 0.02 | 8.12 ± 0.36 | -2.08 ± 1.39 |
> | 64   | 2.11 ± 0.04 | 8.57 ± 0.60 | -1.59 ± 0.39 |
> | 32   | 2.12 ± 0.04 | 8.56 ± 0.64 | -2.36 ± 0.29 |
> | 16   | 2.12 ± 0.04 | 8.65 ± 0.76 | -2.23 ± 0.42 |
> | 8    | 2.12 ± 0.04 | 8.70 ± 0.76 | -1.81 ± 0.99 |
> | 1    | 2.24 ± 0.11 | 9.84 ± 0.79 | -0.48 ± 0.76 |
>
>  **Effect of the test batch size on the calibration performance when using $DATS_w$**
> | Batch size | NLL (↓) | AECE (↓) | ΔLECE (↓) |
> |:-|:-:|:-:|:-:|
> | Full | 2.04 ± 0.02 | 7.88 ± 0.60 | -3.13 ± 0.23 |
> | 128  | 2.08 ± 0.02 | 8.27 ± 0.67 | -2.85 ± 0.51 |
> | 64   | 2.09 ± 0.03 | 8.48 ± 0.64 | -2.72 ± 0.60 |
> | 32   | 2.10 ± 0.03 | 8.48 ± 0.68 | -2.77 ± 0.70 |
> | 16   | 2.10 ± 0.03 | 8.64 ± 0.56 | -2.51 ± 0.80 |
> | 8    | 2.11 ± 0.04 | 8.79 ± 0.64 | -2.86 ± 0.29 |
> | 1    | 2.24 ± 0.11 | 9.84 ± 0.79 | -0.48 ± 0.76 |

---

> ### Author Response · Authors · 2025-11-21
> **Response to Reviewer pqgq (2/3)**
>
> 3. **Limited integration with CIL algorithms**:
>  To demonstrate broader applicability of our approach and its plug-in nature, we followed your suggestion and performed additional experiments considering different families of CL strategies. In particular, we decided to employ the following strategies: GDumb (see response to *Reviewer D1Dg*), Learning without Forgetting (LwF) (knowledge distillation), Elastic Weight Consolidation (EWC) (parameter regularisation), and Retrospective Adversarial Replay (RAR) (more sophisticated memory-based approach). The results demonstrate that our approach can be combined with diverse CL strategies and consistently reduces ECE in all cases.
>
>
>
> **Elastic Weight Consolidation (EWC)**
>
>  | Method | NLL (↓) CIFAR10 | AECE (↓) CIFAR10 | ΔLECE (↓) CIFAR10 | NLL (↓) CIFAR100 | AECE (↓) CIFAR100 | ΔLECE (↓) CIFAR100 | NLL (↓) TinyImageNet | AECE (↓) TinyImageNet | ΔLECE (↓) TinyImageNet |
> |-|:-:|:-:|:-:|:-:|:-:|:-:|:-:|:-:|:-:|
> | Acc | 64.72 ± 2.26 | | | 48.23 ± 0.77 | | | 10.79 ± 0.45 | | |
> | EWC | 2.07 ± 0.35 | 27.08 ± 2.81 | &cross; | 2.63 ± 0.05 | 27.56 ± 0.84 | &cross; | 6.11 ± 0.13 | 54.41 ± 0.93 | &cross; |
> | +DATS | **1.12 ± 0.05** | **8.05 ± 1.84** | -1.12 ± 2.72 | 2.05 ± 0.02 | 7.92 ± 0.56 | -0.82 ± 1.66 | **4.28 ± 0.03** | 12.63 ± 0.46 | **-4.53 ± 0.82** |
> | +DATS_w | 1.13 ± 0.05 | 9.65 ± 2.05 | **-1.35 ± 2.85** | **2.04 ± 0.02** | **7.88 ± 0.60** | **-3.13 ± 0.23** | 4.28 ± 0.02 | **12.44 ± 0.22** | 1.31 ± 0.90 |
>
>
> | Method | NLL (↓) Blood Cell | AECE (↓) Blood Cell | ΔLECE (↓) Blood Cell | NLL (↓) Skin Lesions | AECE (↓) Skin Lesions | ΔLECE (↓) Skin Lesions |
> |-|:-:|:-:|:-:|:-:|:-:|:-:|
> | Acc | 78.63 ± 1.79 | | | 48.83 ± 2.67 | | |
> | EWC | 1.25 ± 0.17 | 15.61 ± 1.68 | &cross; | 1.55 ± 0.11 | 15.04 ± 2.49 | &cross; |
> | +DATS | 0.84 ± 0.21 | 9.28 ± 2.15 | 0.57 ± 0.68 | 1.42 ± 0.04 | 11.55 ± 3.02 | **2.35 ± 3.74** |
> | +DATS_w | **0.79 ± 0.20** | **8.70 ± 1.85** | **0.56 ± 0.67** | **1.41 ± 0.05** | **10.28 ± 1.69** | 2.84 ± 4.48 |
>
>
> **Learning without Forgetting (LwF)**
>
> | Method | NLL (↓) CIFAR10 | AECE (↓) CIFAR10 | ΔLECE (↓) CIFAR10 | NLL (↓) CIFAR100 | AECE (↓) CIFAR100 | ΔLECE (↓) CIFAR100 | NLL (↓) TinyImageNet | AECE (↓) TinyImageNet | ΔLECE (↓) TinyImageNet |
> |-|:-:|:-:|:-:|:-:|:-:|:-:|:-:|:-:|:-:|
> | Acc | 62.19 ± 2.40 | | | 45.13 ± 0.63 | | | 11.89 ± 0.61 | | |
> | LwF | 3.06 ± 0.40 | 28.87 ± 2.65 | &cross; | 2.99 ± 0.09 | 27.38 ± 0.84 | &cross; | 10.74 ± 0.21 | 57.33 ± 1.41 | &cross; |
> | +DATS | 1.29 ± 0.07 | 9.83 ± 4.73 | -0.76 ± 1.12 | **2.26 ± 0.03** | 6.36 ± 0.21 | -1.06 ± 0.48 | **4.58 ± 0.03** | **8.98 ± 0.28** | **1.69 ± 6.79** |
> | +DATS_w | **1.29 ± 0.06** | **9.20 ± 4.22** | **-2.21 ± 0.72** | **2.26 ± 0.03** | **6.29 ± 0.21** | **-3.04 ± 0.47** | 4.61 ± 0.03 | 10.20 ± 0.55 | 5.24 ± 1.52 |
>
> | Method | NLL (↓) Blood Cell | AECE (↓) Blood Cell | ΔLECE (↓) Blood Cell | NLL (↓) Skin Lesions | AECE (↓) Skin Lesions | ΔLECE (↓) Skin Lesions |
> |-|:-:|:-:|:-:|:-:|:-:|:-:|
> | Acc | 80.05 ± 1.63 | | | 47.05 ± 6.23 | | |
> | LwF | 1.09 ± 0.05 | 13.96 ± 1.26 | &cross; | 1.46 ± 0.09 | 15.56 ± 4.12 | &cross; |
> | +DATS | 0.89 ± 0.07 | **11.50 ± 1.37** | 0.70 ± 0.28 | 1.40 ± 0.09 | 13.22 ± 4.71 | **-1.21 ± 1.00** |
> | +DATS_w | **0.83 ± 0.13** | 12.19 ± 1.22 | **0.69 ± 0.28** | **1.38 ± 0.08** | **11.06 ± 2.52** | 0.96 ± 2.91 |
>
>
> **Retrospective Adversarial Replay (RAR)**
>
> | Method | NLL (↓) CIFAR10 | AECE (↓) CIFAR10 | ΔLECE (↓) CIFAR10 | NLL (↓) CIFAR100 | AECE (↓) CIFAR100 | ΔLECE (↓) CIFAR100 | NLL (↓) TinyImageNet | AECE (↓) TinyImageNet | ΔLECE (↓) TinyImageNet |
> |-|:-:|:-:|:-:|:-:|:-:|:-:|:-:|:-:|:-:|
> | Acc | 63.05 ± 1.08 |  |  | 40.73 ± 1.15 |  |  | 20.53 ± 1.45 |  |  |
> | RAR | 1.66 ± 0.12 | 24.48 ± 1.94 | &cross; | 2.90 ± 0.05 | 24.84 ± 0.82 | &cross; | — | — | &cross; |
> | +DATS | **1.08 ± 0.02** | **7.31 ± 0.64** | -0.53 ± 1.04 | **2.45 ± 0.04** | 6.89 ± 0.07 | -1.64 ± 2.38 | — | — | — |
> | +DATS_w | **1.08 ± 0.02** | 8.15 ± 0.38 | **-1.66 ± 1.93** | **2.45 ± 0.04** | **6.27 ± 0.36** | **-4.10 ± 1.43** | — | — | — |
>
>
> | Method | NLL (↓) Blood Cell | AECE (↓) Blood Cell | ΔLECE (↓) Blood Cell | NLL (↓) Skin Lesions | AECE (↓) Skin Lesions | ΔLECE (↓) Skin Lesions |
> |-|:-:|:-:|:-:|:-:|:-:|:-:|
> | Acc | 35.77 ± 12.00 |  |  | 45.46 ± 0.64 |  |  |
> | RAR | 4.42 ± 1.73 | 46.39 ± 10.79 | &cross; | 2.00 ± 0.03 | 26.05 ± 8.65 | &cross; |
> | +DATS | **1.59 ± 0.33** | **16.45 ± 6.81** | **-4.64 ± 4.71** | **1.54 ± 0.13** | 18.51 ± 1.69 | -10.08 ± 8.68 |
> | +DATS_w | 1.62 ± 0.35 | 17.34 ± 5.78 | -3.92 ± 4.28 | **1.54 ± 0.13** | **18.38 ± 1.51** | **-10.42 ± 9.16** |

---

> > ### Author Response · Authors · 2025-11-21
> > **Response to Reviewer pqgq (3/3)**
> >
> > 4. **Novelty boundary**:
> >  We respectfully disagree that the novelty lies mainly in introducing a distance-aware temperature modulation. As mentioned by *Reviewer xTqV*, we identify and substantiate a critical issue of the current post-hoc calibration techniques for class-incremental learning. We then devise an alternative and effective solution that successfully addresses the observed limitations and proves better calibration results in different scenarios.

---

> ### Comment · Reviewer_pqgq · 2025-11-24
> **Official comment**
>
> Thank you for the detailed rebuttal and for the additional experiments. I appreciate the effort you put into addressing our concerns.
>
> There are further questions and comments.
> First, I do not think GDumb should be interpreted as a solution for stabilizing representations on a per-task basis. GDumb mainly demonstrates that one can still achieve good performance by training only a linear classifier on a memory buffer. However, the latent space can still be heavily biased toward the current task. Therefore, it is hard to conclude that the performance gains of "GDumb + DATS" are solely due to improved miscalibration.
>
> Second, the test-time setting where many samples from different tasks are processed together in a single batch reflects only a limited subset of realistic deployment scenarios. In practice, a batch size of 1 is often more representative and thus more critical. In this regime, however, the performance gains appear quite marginal.
>
> Third, it would be helpful to see the corresponding baseline results for DATS when evaluated with batch size 1. Could you report the performance of DATS under this more realistic test-time setting as well?

---

> > ### Author Response · Authors · 2025-11-27
> > **Response to Official comment**
> >
> > Thanks *Reviewer pqgq* for your quick reply. Below is our response to the remaining concerns.
> >
> > 1. We politely disagree with the statement that "the latent space can still be heavily biased toward the current task" in GDumb and think this may be due to a misunderstanding of the subtleties on how GDumb trains the model. GDumb has two main components: 1) a greedy sampler to maintain a *balanced* set of samples from the streaming of tasks, and 2) a classifier model that, when requested, is fully trained (not only the last layer) on the balanced memory only. For this reason, there is no bias toward the last (or any) task as the model is never trained on a specific task but only on a set of balanced samples from all tasks. We therefore believe that the GDumb experiment, as also suggested by *reviewer D1Dg*, is a very neat way to control for bias in the learnt representation.
> >
> > 2. We acknowledge the importance of the evaluation of the model with a test batch size of 1 and its relevance in online or streaming inference scenarios. However, we would like to clarify that our work follows the standard offline class-incremental learning (CIL) setting in which a) the evaluation is typically performed on the full test set or large batches of it (that we have ablated), and b) the model is not required to make decisions on a per-sample basis. To consider such online settings - which have their own large body of literature in the sub-field of online continual learning [1,2,3,4] -  we would need to not only change the procedure at inference time but also comply with specific procedures (e.g., mini-batch streaming, single-pass training, etc) that are specific to online continual learning. For this reason, we believe that evaluating our approach in such specific scenario is beyond the scope of our paper.
> >
> >
> > [1] Aljundi, Rahaf, et al. "Online continual learning with maximal interfered retrieval." Advances in neural information processing systems 32 (2019).
> >
> > [2] He, Jiangpeng, and Fengqing Zhu. "Online continual learning via candidates voting." Proceedings of the IEEE/CVF winter conference on applications of computer vision. 2022.
> >
> > [3] Guo, Yiduo, Bing Liu, and Dongyan Zhao. "Online continual learning through mutual information maximization." International conference on machine learning. PMLR, 2022.
> >
> > [4] Gunasekara, Nuwan, et al. "Survey on online streaming continual learning." Proceedings of the Thirty-Second International Joint Conference on Artificial Intelligence. 2023.

---

### Author Response · Authors · 2025-11-21
**General response to reviewers**

We thank the reviewers for their detailed and thoughtful feedback. We are pleased that the reviewers appreciate the investigation of the underexplored problem of uncertainty calibration in CIL (*Reviewer pqgq*) and the clear identification of critical issues of current post-hoc calibration solutions in this setting (*Reviewer pqgq* and *Reviewer xTqV*). The proposed solution is "aligned with the motivation" (*Reviewer DxNp*) and is deemed "ingenious" (*Reviewer xTqV*), "effective and straighforward" (*Reviewer D1Dg*). The experiments are considered "extensive" (*Reviewer y6Qa*), "diverse and comprehensive" (*Reviewer DxNp*).

We are thankful for the encouraging and constructive comments aimed at improving the quality and soundness of our manuscript. Below we summarize the key points addressed during the rebuttal period:

- **Test-time calibration**:
In response to *Reviewer pqgq* and *Reviewer D1Dg*, we include an analysis of the test batch size (Full, 128, 64, 32, 16, 8, 1) on the calibration performance.

- **Additional CIL baselines**:
Following the request of *Reviewer pqgq*, *Reviewer xTqV*, and *Reviewer DxNp* we now include additional experiments with different families of CIL strategies (LwF, EWC, and RAR). In response to concerns raised by *Reviewer D1Dg* and *Reviewer pqgq* about the bias of the classifier, we also include GDumb.

- **Additional ablation studies**:
We follow the suggestions of *Reviewer DxNp* and now include an analysis of the distance-temperature relationship and ablation studies of the distance criterion (min, max, avg). In response to *Reviewer xTqV*, we propose a new solution (i.e., $DATS_w$) to use a weighted average instead of a threshold for computing the distance score of the samples in the test set (or batch).

---

### Author Response · Authors · 2025-11-21
**Revised version of the paper**

Dear all,

Following the comments and suggestions in the reviews, we uploaded a new version of the paper. The revision includes the following major changes:

1. **Test-time calibration**:
In response to *Reviewer pqgq* and *Reviewer D1Dg*, we include an analysis of the test batch size (Full, 128, 64, 32, 16, 8, 1) on the calibration performance (Appendix A.8 - Table 19).

2. **Additional CIL baselines**:
Following the request of *Reviewer pqgq*, *Reviewer xTqV*, and *Reviewer DxNp* we now include additional experiments with different families of CIL strategies (LwF (Appendix A.6.2 - Tables 9 and 10), EWC (Appendix A.6.3 - Tables 11 and 12), and RAR (Appendix A.6.4 - Tables 13 and 14)). In response to concerns raised by *Reviewer D1Dg* and *Reviewer pqgq* about the bias of the classifier, we also include GDumb (Appendix A.6.1 - Tables 7 and 8).

3. **Additional ablation studies**:
We follow the suggestions of *Reviewer DxNp* and now include an ablation study of the distance criterion (min, max, avg) (Appendix A.8 - Table 20). In response to *Reviewer xTqV*, we propose a new solution (i.e., $DATS_w$) to use a weighted average instead of a threshold for computing the distance score of the test batch (Appendix A.2).

---

### Author Response · Authors · 2025-12-03
**Rebuttal Summary for Area Chair (1/3)**

**Paper summary:** In our manuscript, we tackle the challenges of **uncertainty calibration in class-incremental learning (CIL)**, a **largely under-explored problem**. By investigating the behaviour of models trained in CIL regimes, we identify a critical limitation of current CIL post-hoc calibration techniques: they primarily focus on the data to use for calibration and **overlook task variability** – leading to large fluctuations in expected calibration error (ECE). For this reason, we depart from such task-agnostic, data-centric view and are the **first to propose an adaptive, task-aware calibration method** for CL. To instill task-awareness into the calibration phase, we propose DATS, an elegant solution that, via a combination of prototype-based distance estimation and distance-aware calibration, is able to **tune the temperature without prior knowledge** about the considered task. Our method demonstrates **consistent and significant effectiveness** across diverse settings while maintaining the simplicity and plug-and-play nature that makes it practically valuable.

---

> ### Author Response · Authors · 2025-12-03
> **Rebuttal Summary for Area Chair (2/3)**
>
> In their constructive and insightful comments, reviewers have raised four major concerns that were commonly shared across them. We have addressed those concerns in full, as summarized below:
>
> - **Limited integration with CIL baselines (Reviewer pqgq, Reviewer xTqV, and Reviewer DxNp)**: we have now included additional experiments with different families of CIL strategies, namely Learning without Forgetting (LwF; knowledge distillation), Elastic Weight Consolidation (EWC; parameter regularisation), Retrospective Adversarial Replay (RAR; more sophisticated memory-based approach), and GDumb (bias-free). *Reviewer DxNp* stated that "the additional experiments demonstrating the plug-and-play capability are sufficiently convincing". *Reviewer xTqV* acknowledged that we have addressed their concerns. *Reviewer pqgq*, in their follow-up comment, did not mention this point as a remaining concern.
> - **Assumption of batched test data (Reviewer pqgq and Reviewer D1Dg)**: We conducted additional experiments which show that DATS is able to perform well across a wide range of test batch sizes, including the edge case where only a single sample is available (i.e., batch size = 1). *Reviewer pqgq* still had concerns about our batched assumption. However, we would like to politely highlight that this stems from a methodological preference rather than a technical flaw. Being DATS rooted in the literature for offline CIL, a) the evaluation is typically performed on the full test set or large batches of it (that we have ablated), and b) the model is not required to make decisions on a per-sample basis. To consider such online settings - which have their own large body of literature in the sub-field of online continual learning [1,2,3,4] - we would need to not only change the procedure at inference time but also comply with specific procedures (e.g., mini-batch streaming, single-pass training, etc) that are specific to online continual learning. For this reason, we believe that evaluating our approach in such specific scenario is beyond the scope of our paper.
> - **Representation bias concern (Reviewer pqgq and Reviewer D1Dg)**: Two reviewers raised concerns on whether miscalibration stems primarily from classifier representations being biased toward recent tasks rather than genuine task-recency effects. Following the explicit suggestion of *Reviewer D1Dg*, we addressed this by incorporating GDumb as a baseline without representation bias. DATS still consistently reduces calibration error on top of GDumb across all datasets. This demonstrates that our distance-aware temperature scaling method provides benefits that go beyond what can be achieved through bias mitigation alone. *Reviewer pqgq* was still unconvinced about the use of GDumb since, in their opinion, *"the latent space can still be heavily biased toward the current task"*. However, GDumb trains exclusively on a *balanced* memory buffer (never on task-specific data), eliminating any bias toward the last or any particular task. We believe *Reviewer pqgq*'s comment on the bias of GDumb was based on a misunderstanding as they state that GDumb trains "only a linear classifier on a memory buffer". We clarified that GDumb does not merely train the linear head, but actually trains the entire classifier. Therefore, we believe that the GDumb experiment, as also suggested by reviewer D1Dg, is a very neat way to control for bias in the learnt representations.
>
> - **Distance-temperature relationship (Reviewer DxNp and Reviewer xTqV)**: We provided explanations and empirical evidence of the distance-temperature correlation. *Reviewer DxNp* was not convinced by our first response and encouraged us to perform a comparison where distances are replaced with fixed constants as a way to "significantly strengthen the technical soundness". We first clarified that we presented a concrete example from CIFAR100 showing a high correlation between distance and temperature. The results are consistent with prior analyses on calibration under distribution shift [5,6], where over-confidence increases when inputs fall outside recently trained regions, and justify the use of higher temperatures for farther regions. In addition, we followed the specific suggestion of distances as fixed constants and replaced actual distance scores with a constant value during optimization. The results confirm that distance scores carry intrinsic information that is practically essential for effective calibration, not just providing an additional degree of freedom. We did not receive a further response before comments were closed, but are convinced to have fully addressed this concern.

---

> > ### Author Response · Authors · 2025-12-03
> > **Rebuttal Summary for Area Chair (3/3)**
> >
> > In addition, there were reviewer-specific concerns that we addressed as summarized below:
> >
> > 1. **Reviewer pqgq** (all addressed, no final feedback)
> >     - **Novelty boundary**: We clarified that the novelty does not lie mainly in introducing a distance-aware temperature modulation. In the follow-up response, *Reviewer pqgq* did not mention this point as a remaining concern.
> >
> > 2. **Reviewer D1Dg** (all addressed, no interaction)
> >     - **Lack of classification performance metrics**: We clarified that, being a post-hoc calibration method based on accuracy-preserving temperature scaling, the classification performance is not affected by our method and is reported in parentheses next to the name of the considered dataset.
> >
> > 3. **Reviewer xTqV** (all addressed, no other concerns)
> >     - **Stability and reliability of older class prototypes**: we performed additional experiments that demonstrate that our approach can perform well also in low-accuracy regimes.
> >     - **Limitations of fixed-threshold mechanism**: we addressed this by introducing $DATS_w$, a threshold-free variant which achieves comparable or better performance than the original DATS across all datasets and CIL strategies, while completely avoiding the need for explicit threshold tuning.
> >     - **Calibration buffer sensitivity**: We clarified that an ablation study was already present in the paper. Results show that changing the calibration buffer size has only a minor impact on performance, demonstrating robustness across memory constraints.
> >
> > 4. **Reviewer y6Qa** (all addressed, no interaction)
> >     - **Implications of perfect calibration**: We provided the implications of perfect calibration based upon recent theoretical work on proper calibration errors [7].
> >     - **Comparison with [8]**: We clarified that a comprehensive comparison with [8] is already present in the manuscript.
> >
> > 5. **Reviewer DxNp** (all addressed, no final feedback)
> >     - **Theoretical justification for prototype-based approach**: after the first response based on empirical observations summarised in Figures 3 and 4, *Reviewer DxNp* asked for more theoretical clarifications. We then offered solid theoretical justifications based on shared-covariance assumptions and alignment of our methodological choice with temperature scaling expressive capacity.
> >     - **Distance definition is heuristic**: We ablated the choice of the distance definition and clarified that we use the minimum distance between class prototypes as a proxy of the closeness between classes. The justification was deemed reasonable and convincing, but tightly linked with **Theoretical justification for prototype-based approach** for which we then provided a more thorough theoretical justification.
> >
> >
> > We appreciate the time and effort invested in assessing our work. We also sincerely appreciate the constructive comments and insightful questions from reviewers, which substantially improved the quality and clarity of our manuscript. We believe that the revised manuscript now addresses all concerns.
> >
> >
> >
> > [1] Aljundi, Rahaf, et al. "Online continual learning with maximal interfered retrieval." Advances in neural information processing systems 32 (2019).
> >
> > [2] He, Jiangpeng, and Fengqing Zhu. "Online continual learning via candidates voting." Proceedings of the IEEE/CVF winter conference on applications of computer vision. 2022.
> >
> > [3] Guo, Yiduo, Bing Liu, and Dongyan Zhao. "Online continual learning through mutual information maximization." International conference on machine learning. PMLR, 2022.
> >
> > [4] Gunasekara, Nuwan, et al. "Survey on online streaming continual learning." Proceedings of the Thirty-Second International Joint Conference on Artificial Intelligence. 2023.
> >
> > [5] Ovadia, Yaniv, et al. "Can you trust your model's uncertainty? evaluating predictive uncertainty under dataset shift." Advances in neural information processing systems 32 (2019).
> >
> > [6] Minderer, Matthias, et al. "Revisiting the calibration of modern neural networks." Advances in neural information processing systems 34 (2021): 15682-15694.
> >
> > [7] Gruber, Sebastian, and Florian Buettner. "Better uncertainty calibration via proper scores for classification and beyond." Advances in Neural Information Processing Systems 35 (2022): 8618-8632.
> >
> > [8] Li, Lanpei, et al. "Calibration of continual learning models." Proceedings of the IEEE/CVF Conference on Computer Vision and Pattern Recognition. 2024.

---

### Meta-Review · Area_Chair_sSLc · 2026-01-02

**Summary:**

Reviewers appreciated the straightforwardness of the proposed method to reduce expected calibration error in continual learning. At the same time, most of the reviewers had common concerns over misattribution of the effect, limited empirical analysis in conjunction with existing continual learning algorithms, insufficient theoretical motivation, and unrealistic set-up assumptions. Extensive responses have been provided and fortunately, some discussion occurred prior to this year's freeze. Throughout these discussions it appears as it many concerns remain unaddressed, leading to two reviewers being reluctant to raise their scores. Going through the remaining reviews, discussion, and revision, the AC believes that various concerns have been in parts addressed, but also sees the main critical points to remain largely open. The AC suggests a more careful and thorough revision, following below suggestions, and at present recommends to reject the paper in its present form.

**Reviewer Concerns:**

Among the many raised points, reviewers had shared concerns over 1. the potential misattribution of the effect (due to e.g. classifier bias), 2. limited experimental scope and inclusion of prominent continual learning approaches, 3. lack of theoretical justification, 4. limited considered setting.

Following the rebuttal, discussion, and revision, the AC believes only the first two points to have been addressed partially, whereas the latter two points to remain almost fully open. Regarding the misattribution of the effect, experiments using GDUMB have been included. Whereas some evidence has thus been gathered, the experiments also show that there is a different additional narrative at play that should be investigated much more carefully. As pointed out by reviewers, this is a critical aspects that needs to be done justice in the paper, beyond a single experiment with GDUMB. Regarding the second and fourth point, the AC believes the additional experimentation should be included in the main body, rather than following a post-hoc addition to some appendix section. The AC also agrees with the reviewers that the setting is somewhat limited and potentially unrealistic, limiting the novelty and potential significance of the work. The rebuttal has made the choice to frame this point as "out of scope", but more consideration should be given to this aspect of practicability. Finally, the lack of theoretical justification is addressed in the rebuttal by stating that some properties regarding prototypes are true if one assumes everything is distributed in a Gaussian matter. The AC believes this is an insufficient theoretical justification of the method, as it simply relies on a different very strong assumption (that is often not true without ensuring it with methods such as Bayesian inference). The theoretical part here is rather hand-wavy. As such, many of the concerns, especially those by the more critical reviewers remain insufficiently addressed.

**Reviewer Scores:**

Reviewer scores were 2,2,4,4,6, with one reviewing giving a 4 stating a confidence of 1 and the review with the score of 6 lacking any substance (the review is unfortunately short and shallow). In contrast, the other three reviewers gave extensive feedback, which has been rebutted and in two cases discussed prior to the freeze. In the two cases where discussion could happen, some acknowledgements by the reviewers were made regarding a part of their concerns being clarified. Yet, concerns also remained or the resolving of questions still resulted in the reviewer to actively retain a recommendation to reject the paper in its present form. Extrapolating to a longer discussion that could not happen due to the OpenReview incident, the AC believes that the latter is due to the fact that some of the criticism is not easily addressable through a rebuttal. In particular, the theoretical justification, the inadequate embedding and comparison to continual learning methods, and the limited experimental set-up are not easily overcome through a rebuttal alone. Following this reasoning, the AC believes it would have been unlikely for reviewers to have raised their scores substantially. Two reviewers actively made the decision to retain their reject recommendations, and one reviewer who gave a 2 had major unresolved critique. Although one reviewer gave a 6, their review is unfortunately very shallow and not helpful for the decision making process. As such, the overall reviews lean to the negative side and a much longer discussion period would have been unlikely to turn this around fully.

---

### Decision · Program_Chairs · 2026-01-26

Reject